# PACSIN1 is indispensable for amphisome-lysosome fusion during basal autophagy and subsets of selective autophagy

Yukako Oe[1], Keita Kakuda[2], Shin-ichiro Yoshimura[3], Naohiro Hara[4], Junya Hasegawa[5], Seigo Terawaki[6], Yasuyoshi Kimura[2], Kensuke Ikenaka[2], Shiro Suetsugu[7,8,9], Hideki Mochizuki[2], Tamotsu Yoshimori[1,4,10]*, Shuhei Nakamura[1,4,11]*

1 Laboratory of Intracellular Membrane Dynamics, Graduate School of Frontier Biosciences, Osaka University, Osaka, Japan, 2 Department of Neurology, Graduate School of Medicine, Osaka University, Osaka, Japan, 3 Department of Cell Biology, Graduate School of Medicine, Osaka University, Osaka, Japan, 4 Department of Genetics, Graduate School of Medicine, Osaka University, Osaka, Japan, 5 Department of Biochemical Pathophysiology, Medical Research Institute, Tokyo Medical and Dental University, Tokyo, Japan, 6 Department of Molecular and Genetic Medicine, Kawasaki Medical School, Okayama, Japan, 7 Division of Biological Science, Graduate School of Science and Technology, Nara Institute of Science and Technology, Ikoma, Japan, 8 Data Science Center, Nara Institute of Science and Technology, Ikoma, Japan, 9 Center for Digital Green-Innovation, Nara Institute of Science and Technology, Ikoma, Japan, 10 Integrated Frontier Research for Medical Science Division, Institute for Open and Transdisciplinary Research Initiatives (OTRI), Osaka University, Osaka, Japan, 11 Institute for Advanced Co-Creation Studies, Osaka University, Osaka, Japan

* tamyoshi@fbs.osaka-u.ac.jp (TY); shuhei.nakamura@fbs.osaka-u.ac.jp (SN)

**Data Availability Statement:** All relevant data are within the manuscript and its Supporting Information files.

## Abstract

Autophagy is an indispensable process that degrades cytoplasmic materials to maintain cellular homeostasis. During autophagy, double-membrane autophagosomes surround cytoplasmic materials and either fuse with endosomes (called amphisomes) and then lysosomes, or directly fuse with lysosomes, in both cases generating autolysosomes that degrade their contents by lysosomal hydrolases. However, it remains unclear if there are specific mechanisms and/or conditions which distinguish these alternate routes. Here, we identified PACSIN1 as a novel autophagy regulator. *PACSIN1* deletion markedly decreased autophagic activity under basal nutrient-rich conditions but not starvation conditions, and led to amphisome accumulation as demonstrated by electron microscopic and co-localization analysis, indicating inhibition of lysosome fusion. PACSIN1 interacted with SNAP29, an autophagic SNARE, and was required for proper assembly of the STX17 and YKT6 complexes. Moreover, PACSIN1 was required for lysophagy, aggrephagy but not mitophagy, suggesting cargo-specific fusion mechanisms. In *C. elegans*, deletion of *sdpn-1*, a homolog of *PACSINs*, inhibited basal autophagy and impaired clearance of aggregated protein, implying a conserved role of PACSIN1. Taken together, our results demonstrate the amphisome-lysosome fusion process is preferentially regulated in response to nutrient state and stress, and PACSIN1 is a key to specificity during autophagy.

**Funding:** T.Y. is supported by JST CREST (grant no. JPMJCR17H6), AMED (grant no. JP21gm5010001), and the Takeda Science Foundation. S.N. is supported by AMED-PRIME (20gm6110003h0004), MEXT KAKENHI, a Grant-in-Aid for Transformative Research Areas B (21H05145), JSPS KAKENHI (21H02428, 19K22429), the Senri Life Science Foundation, the Takeda Science Foundation, the Nakajima Foundation, the MSD Life Science Foundation, the Astellas Foundation for Research on Metabolic Disorders, the Mochida Memorial Foundation for Medical and Pharmaceutical Research, the Uehara Memorial Foundation, the NOVARTIS Foundation (Japan) for the Promotion of Science and the Mitsubishi Foundation, Research Grants in the Natural Sciences. S.S. is supported by JSPS KAKENHI 20H03252. The funders had no role in study design, data collection and analysis, decision to publish, or preparation of the manuscript.

**Competing interests:** I have read the journal's policy and the authors (T.Y., S.N.) of this manuscript have the following competing interests: AutoPhagyGO. The other authors have declared that no competing interest exist.

## Author summary

Autophagy is an evolutionally conserved cytoplasmic degradation system in which double membrane structure called autophagosomes sequester several cytoplasmic materials and then transport to lysosomes for degradation. Previous studies mainly based on electron microscopy indicates autophagosomes either fuse with lysosomes directly or fuse with endosomes/MVB (Multi Vesicular Body), producing amphisomes then fuse with lysosomes. However, it remains unknown how these processes are regulated and the physiological relevance of these two routes due to lack of key molecules involved in either of two routes precludes the detailed characterization. In the current study, we identified PACSIN1 as a novel regulator of autophagy, whose function is essential for amphisome-lysosome fusion process specifically. Through the analysis of PACSIN1, we revealed that PACSIN1-dependent fusion process via amphisomes is required for basal autophagy and subsets of selective autophagy, suggesting that two autophagic routes are utilized depending on the context and/or cargos.

## Introduction

Macroautophagy (hereafter referred to as autophagy) is a highly conserved intracellular bulk degradation system. Autophagy acts as a quality-control system by promoting the clearance of long-lived proteins, damaged organelles, and aggregated proteins [1]. These functions prevent various human diseases such as neuronal degeneration, hepatic disorders, and muscle atrophy [2,3]. During autophagy, de novo double-membrane structures referred to as autophagosomes engulf cytosolic components. The sequestered materials are transported to lysosomes called autolysosomes, and are degraded by lysosomal acid hydrolases. There are two different paths by which the cargo is transported; the autophagosome may directly fuse with a lysosome, or it may fuse with an endosome, known as an amphisome, and then fuses with lysosome [4–6]. However, the molecular mechanism underlying these two pathways respectively is not well understood. Moreover, it remains unclear if specific conditions preferentially result in either of these routes.

Autophagosome-endo/lysosome fusion is mediated by SNAREs, various tethering factors, and Rabs [7,8]. Membrane fusion is driven by SNAREs, and two SNARE complexes, namely STX17-SNAP29-VAMP8 [9] and YKT6-SNAP29-STX7 [10], mediate fusion between the autophagosome-endo/lysosome. Some tethering factors such as PLEKHM1, the HOPS complex, and EPG5, promote fusion between specific vesicles and assemble or stabilize SNARE complexes [11–15]. Rab proteins localize on specific membranes and determine the recruitment of tethering factors [16,17]. Accumulating evidence has revealed the core molecular machinery involved in the autophagosome-endo/lysosome fusion process. However, it remains poorly understood how components of this machinery work cooperatively and how they function properly, as well as if they are involved in both of the autophagosome fusion paths.

PACSINs are a family of cytoplasmic phosphoproteins that play a crucial role in vesicle formation and transport. They contain a highly conserved N-terminal Fes/CIP4 homology-BAR (F-BAR) and a C-terminal SH3 domain. The F-BAR domain directly binds to lipid membranes and drives membrane curvature and tubulation. The SH3 domain is required for protein-protein interactions, and functions as an adaptor to recruit these proteins [18,19]. PACSINs regulate intracellular vesicle trafficking, cytoskeletal rearrangement, caveolar biogenesis, neuronal

development, and cell migration [20–23]. However, it is largely unknown whether PACSINs participate in the autophagic machinery.

Here we demonstrated that PACSIN1 regulates amphisome-lysosome fusion in basal and subsets of selective autophagy by promoting the formation of SNARE complexes. Moreover, our findings showed that the preferential fusion process path depends on nutrient status and different stresses.

## Results

### Loss of *PACSIN1* impairs autophagic activity

Previous studies revealed that PACSIN family proteins bind to lipid membranes via their F-BAR domains and promote membrane curvature, thereby contributing to membrane nucleation, *e.g.*, the formation of caveolae and synaptic vesicles [22]. However, PACSIN2 and PACSIN3 had been shown to the caveolae biogenesis, while PACSIN1 was not suggested to be localized at caveolae [23]. In addition, PACSINs act as adaptors via the SH3 domain to recruit their binding partners to the membrane, and facilitate spatial and temporal regulation. Autophagy is a complex process in which a flat membrane, called an isolation membrane, acquires curvature to form an autophagosome. Therefore, we speculated that PACSINs are necessary for autophagy due to their membrane-binding ability, especially in terms of achieving membrane curvature or recruiting factors that are required for the autophagic process. To test this idea, we established KO HeLa cells for all *PACSIN* family genes using the CRISPR-Cas9 system (S1A and S1B Fig). To examine the role of PACSINs in autophagy, we first performed an LC3 flux assay to measure autophagic activity using Bafilomycin A1 (Baf.A1), which prevents autophagosome-lysosome fusion or lysosomal degradation by inhibiting the lysosomal proton pump V-ATPase [24,25]. LC3 flux was calculated by subtracting the LC3-II value in the absence of Baf.A1 from that in the presence of Baf.A1 in nutrient-rich or starvation medium both in wild-type (WT) and *PACSINs* KO cells. We found that two independent *PACSIN1* KO cells in nutrient-rich condition already exhibited increased LC3-II levels compared with WT cells (Figs 1A and S1C, comparison between lane 1 and 5, respectively), and did not further increase after the treatment with Baf.A1 (Figs 1A and S1C, comparison between lane 5 and 6, respectively), suggesting that autophagy activity is impaired in *PACSIN1 KO* cells. Indeed, in this condition LC3 flux was decreased in *PACSIN1* KO cells compared to WT cells (Fig 1B). We confirmed that *PACSIN1* KO cells do not show obvious growth defect compared to WT cells (S1D Fig). Surprisingly, under starvation conditions, LC3 flux in *PACSIN1* KO cells was comparable to WT cells. These results suggest that autophagic activity is decreased in *PACSIN1* KO cells only under the nutrient-rich condition, namely basal autophagy. By contrast, depletion of *PACSIN2* and *PACSIN3*, which are other paralogs of *PACSINs*, did not alter autophagic activity (S1E and S1F Fig). Therefore, of three PACSIN family proteins, only PACSIN1 affected autophagic activity. *PACSIN1* KO cells showed accumulation of p62, which is a substrate for autophagic degradation and impaired autophagic activity based on p62 flux which was calculated in same way of LC3 flux only in the nutrient-rich condition (Fig 1C and 1D). This is consistent with our finding that basal autophagic activity was suppressed in *PACSIN1* KO cells. Previous report indicates that PACSIN1 regulates the trafficking of AMPA receptor together with PICK1 [26, 27]. However, we found that PICK1 knockdown using two independent siRNAs do not affect basal autophagy as revealed by LC3 flux assay suggesting that a role of PACSIN1 in autophagy can be distinguished from AMPA receptor trafficking (S2 Fig). An immunostaining assay showed accumulation of LC3 puncta in *PACSIN1* KO cells (Fig 1E and 1F). Knockdown of ATG13, which plays an early role in autophagy, suppressed LC3-II formation both in WT cells and *PACSIN1* KO cells (Fig 1G), suggesting that *PACSIN1* deletion

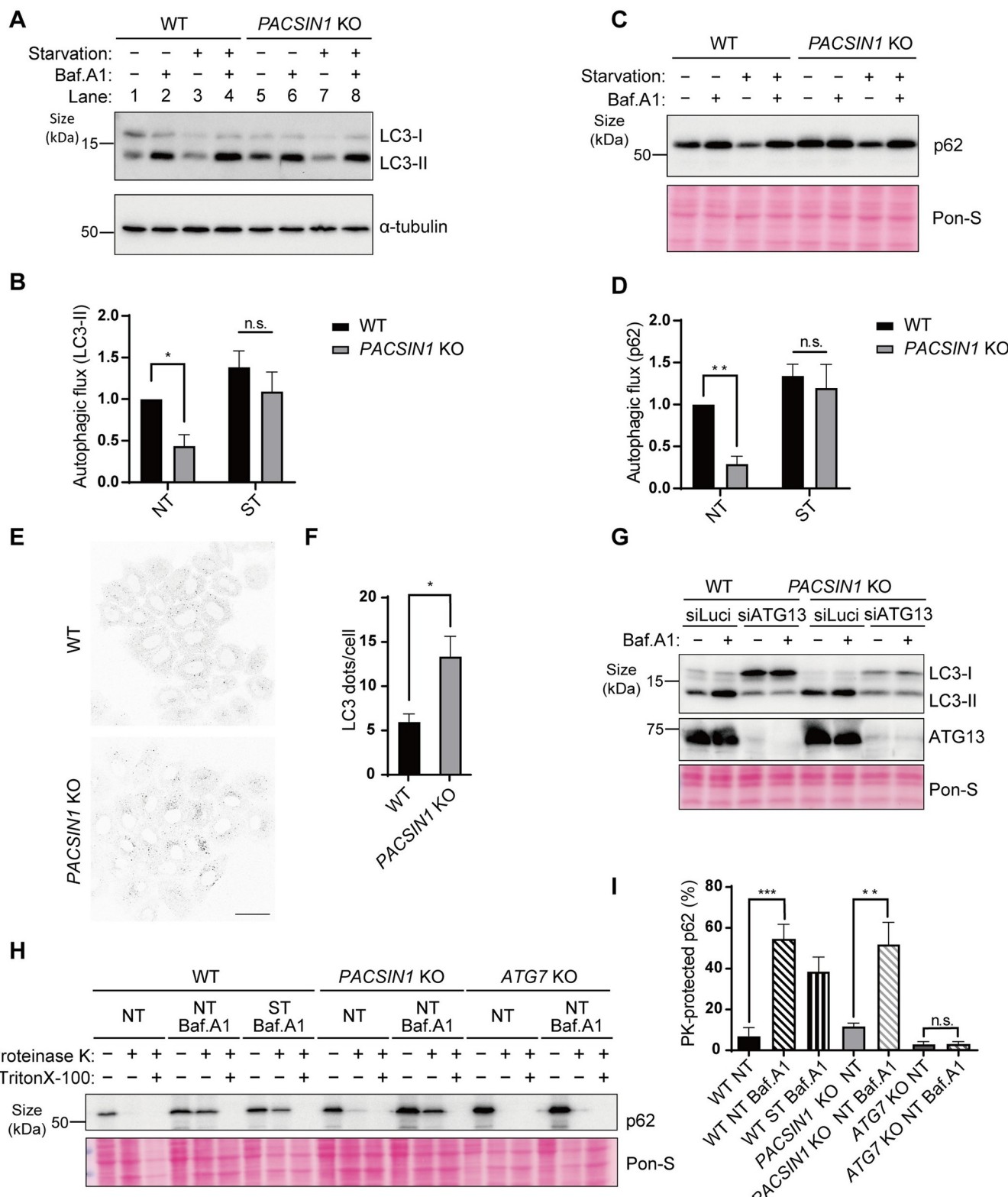

**Fig 1. PACSIN1 is essential for autophagy.** (A) WT or *PACSIN1* clone #1 KO HeLa cells were cultured for 2 h in growth medium (DMEM, NT) or starvation medium (EBSS, ST) with 125 nM bafilomycin A1 (Baf.A1) and then analyzed by immunoblot using anti-LC3 and anti–α-tubulin antibodies. (B) The quantification of LC3 flux is expressed as mean ± s.e.m (n = 3). n.s.; not significant, *$p < 0.05$ (multiple t-test). (C) WT or *PACSIN1* KO HeLa cells were cultured for 6 h in growth medium (NT) or starvation medium (ST) with 125 nM Baf.A1 and then analyzed by immunoblot using anti-p62. Pon-S, Ponceau S

staining. (D) The quantification of p62 flux is expressed as mean ± s.e.m (n = 3). n.s.; not significant, $^{**}p < 0.01$ (multiple t-test). (E) WT or *PACSIN1* KO HeLa cells were cultured in growth medium. Cells were fixed and stained with anti-LC3 antibodies. Scale bars, 40 μm. (F) The numbers of LC3 dots normalized per cell were counted by Fiji, mean ± s.e.m.; More than 200 cells were analyzed per condition in each experiment (n = 4). $^*p < 0.05$ (two-tailed, unpaired t-test). (G) WT or *PACSIN1* KO HeLa cells treated with siLuciferase or siATG13 were cultured for 2 h in growth medium with or without 125 nM Baf.A1, then analyzed by immunoblot using anti-LC3 and anti-ATG13 antibodies. (H) WT or *PACSIN1* KO or *ATG7* KO HeLa cells were cultured for 8 h in growth medium or starvation medium with or without 250 nM Baf.A1. Cells were homogenized and the lysates were treated with proteinase K with or without Triton X-100, and then analyzed by immunoblot using anti-p62 antibodies. (I) Ratio of p62 with/without proteinase K. Mean ± s.e.m (n = 3). n.s.; not significant, $^{**}p < 0.01$, $^{***}p < 0.001$ (one-way ANOVA with Tukey's multiple comparisons test).

caused accumulation of autophagosomes. It is possible that *PACSIN1* deletion failed to generate intact autophagosomes. Therefore, we used a proteinase K protection assay to determine whether or not the accumulated autophagosomes were closed. In this assay, p62 in completely closed autophagosomes, but not in unclosed ones, was protected from degradation by proteinase K. To accumulate enough autophagosomes for this protection assay, we treated cells with Baf.A1 for long time (8 h). This assay showed that p62 escaped from degradation by proteinase K in both WT and *PACSIN1* KO cells, whereas *ATG7* KO cells, in which autophagosomes are defective, showed effective degradation of p62 (Fig 1H and 1I). These findings indicate that enclosed autophagosomes are accumulated in *PACSIN1* KO cells under nutrient-rich conditions.

## *PACSIN1* deletion disrupts a late step of autophagy, but not lysosomal function

*PACSIN1* deletion led to a reduction of autophagic activity, whereas autophagosome formation was normal. Therefore, it is conceivable that the impairment of autophagic activity is caused by reduced lysosomal integrity or a defect in the autophagosome-lysosome fusion process. To test these possibilities, we first assessed lysosomal acidity using Lysotracker, an acidotropic fluorescent dye. After treatment with Baf.A1, which inhibits the lysosomal proton pump V-ATPase, the intensity of Lysotracker was dramatically decreased in WT cells. In the starvation condition, WT cells showed increased Lysotracker intensity due to enhanced lysosomal activity. Importantly, in the nutrient-rich condition, *PACSIN1* KO cells showed higher-intensity Lysotracker staining than WT cells (Fig 2A and 2B). We also measured lysosomal enzyme activity using the Magic Red cathepsin B assay, which reflects the enzymatic activity of its substrate. Although the Magic Red intensity varies among cells both in WT and *PACSIN1* KO cells, the average intensity in *PACSIN1* KO cells was comparable with WT cells (Figs 2C and 2D and S3). These results suggest that the impairment of autophagy observed in *PACSIN1* KO cells is not due to reduced lysosomal function.

Next, we performed a tf-LC3 assay to determine if PACSIN1 contributes to the autophagosome-lysosome fusion process. tf-LC3 is a tandem RFP-GFP–tagged LC3. When autophagosomes fuse with lysosomes, GFP fluorescence on autophagosomes is quenched due to lysosomal acidity, while RFP fluorescence remains stable. Namely, RFP- and GFP-positive structures indicate autophagosomes, while RFP-positive but GFP-negative structures represent autolysosomes. This assay showed that *PACSIN1* KO cells had more RFP- and GFP- positive autophagosomes than WT cells (Fig 2E and 2F). These findings indicate that *PACSIN1* deletion impairs the autophagosome-lysosome fusion process, thereby resulting in the accumulation of autophagosomes.

## *PACSIN1* deletion causes amphisome accumulation

To further investigate the detailed structure of the accumulated autophagosomes in *PACSIN1* KO cells, we performed an electron microscopy analysis. We also included samples treated

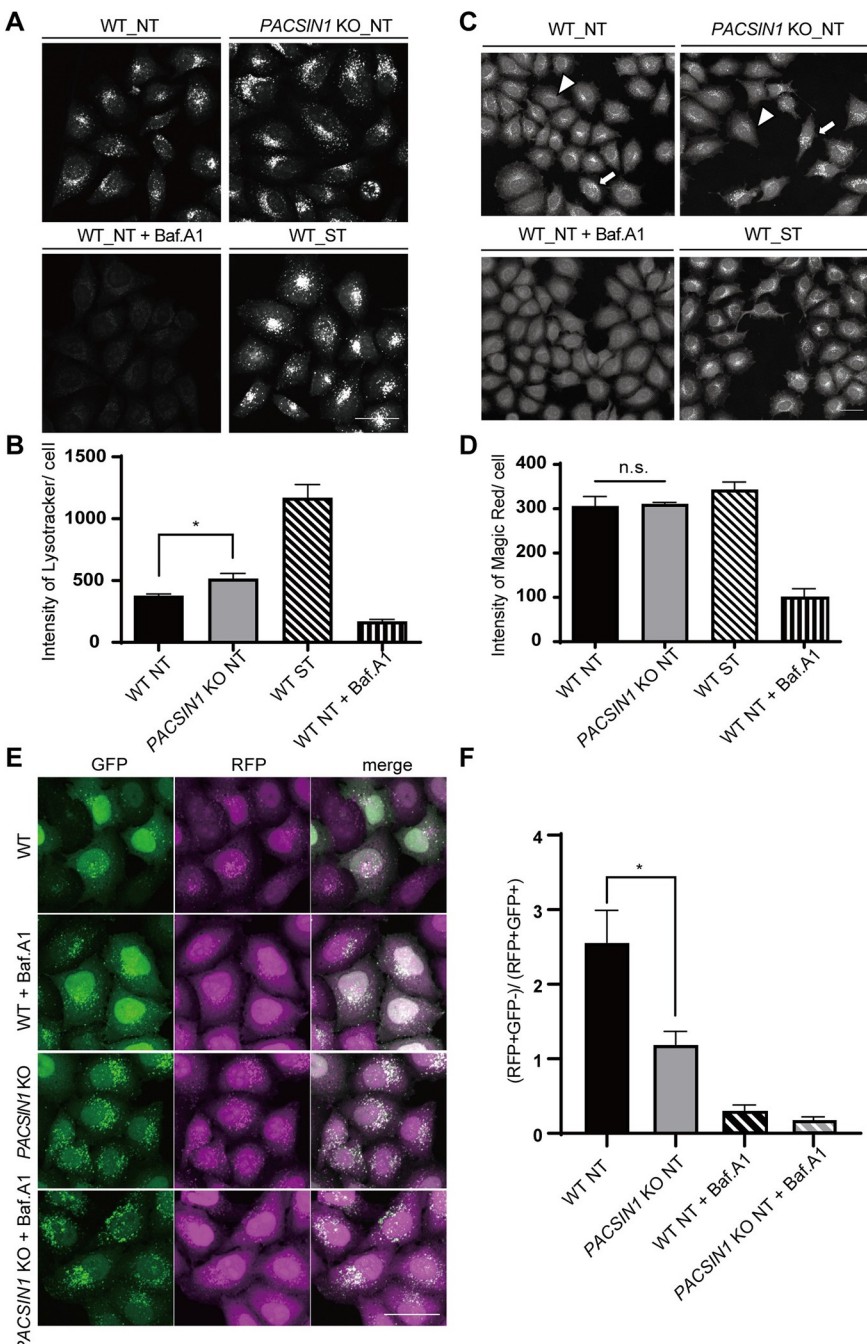

**Fig 2. PACSIN1 is required for the autophagosome-lysosome fusion process but not for lysosomal function.** (A) WT or *PACSIN1* KO HeLa cells were cultured for 2 h in growth medium or starvation medium containing 50 nM Lysotracker Red with or without 125 nM Baf.A1. Cells were fixed and then analyzed using CQ1 software. Scale bars, 40 μm. (B) Quantified and normalized Lysotracker mean intensity per cell, mean ± s.e.m. More than 100 cells were analyzed per condition in each experiment (n = 3). *$p < 0.05$ (two-tailed, unpaired t-test). (C) WT or *PACSIN1* KO HeLa cells were cultured for 2 h in growth medium or EBSS with or without 125 nM Baf.A1 and treated with Magic Red. After fixation, cells were analyzed using CQ1 software. The cell exhibited relatively high and low intensity of Magic Red were indicated by arrows and arrowheads, respectively. Scale bars, 40 μm. (D) Quantified Magic Red mean intensity normalized per cell, mean ± s.e.m. More than 100 cells were analyzed per condition in each experiment (n = 3). n.s.; not significant (two-tailed, unpaired t-test). (E) WT or *PACSIN1* KO HeLa cells stably expressing tf-LC3 were cultured for 2 h in growth medium with or without 250 nM Baf.A1. After fixation, cells were analyzed using CQ1 software. Scale bars, 40 μm. (F) The numbers of RFP+GFP−/RFP+GFP+ dots were counted, mean ± s.e.m. More than 100 cells were analyzed per condition in each experiment (n = 3). *$p < 0.05$ (two-tailed, unpaired t-test).

with Baf.A1 for long time (6 h) to accumulate and observe autophagosomes. Contrary to our expectation that many autophagosomes would accumulate in *PACSIN1* KO cells, we found that *PACSIN1* deletion led to the accumulation of structures resembling amphisomes (Figs 3A, arrows, and S4A) [4,28,29]. In addition, immuno-electron microscopy analysis showed that immunogold particles recognizing LC3 were localized on these amphisome-like structures (S4B Fig). To confirm whether these structures were indeed amphisomes, we examined the co-localization between LC3 and the late endosomal marker CD63 or the lysosomal marker Lamp1. *PACSIN1* KO cells and WT cells showed a similar co-localization rate of LC3 with Lamp1 (S4C and S4D Fig). By contrast, the co-localization rate of LC3 with CD63 was higher in *PACSIN1* KO cells than in WT cells (Fig 3B and 3C). These results suggested that *PACSIN1* deletion caused amphisome accumulation. Furthermore, to determine whether this accumulation was due to a defect in the flow of transport via autophagosomes/amphisomes-lysosomes, we treated cells with E-64-d and pepstatin A, both of which inhibit lysosomal protease. An increase in the co-localization rate after inhibitor treatment should represent increased flow of transport to lysosomes. Co-localization of LC3 with Lamp1 was increased in both WT and *PACSIN1* KO cells compared with non-treated cells, and the increment of the co-localization in WT cells was greater than that in *PACSIN1* KO cells (S4C and S4D Fig). By contrast, although the co-localization rates of LC3 and CD63 were significantly increased after inhibitor treatment in WT cells, it was not altered in *PACSIN1* KO cells (Fig 3B and 3C). These observations together suggested that *PACSIN1* deletion resulted in the blockage of fusion between amphisomes and lysosomes.

## PACSIN1 localizes on autophagosomes/amphisomes and lysosomes

To address the mechanism by which PACSIN1 participates in the autophagosome/amphisome-lysosome fusion process, we first examined the localization of PACSIN1. EGFP-PACSIN1 showed a cytoplasmic pattern that could mask the punctate structures of PACSIN1. Therefore, cells were pre-treated with saponin to remove the cytoplasmic signals. After saponin treatment, EGFP-PACSIN1 exhibited punctate structures, and most of them co-localized with Lamp1 and partially with LC3 (Fig 4), suggesting that PACSIN1 localizes on autophagosomes/amphisomes and lysosomes.

## PACSIN1 is required for assembly of autophagic SNARE complexes

Our results indicate that *PACSIN1* deletion reduces autophagic activity and results in the accumulation of amphisomes. Therefore, we speculated that *PACSIN1* deletion causes a defect in amphisome-lysosome fusion, thereby leading to abnormal amphisome accumulation. Previous reports showed that autophagosome and endo/lysosome fusion is mediated by two SNAREs complexes: STX17(Qa)-SNAP29(Qbc)-VAMP7/8(R) [9] and YKT6(R)-SNAP29(Qbc)-STX7 (Qa) [10]. Thus, we examined the formation of these complexes in *PACSIN1* KO cells. We performed immunoprecipitation of GFP-STX17 using a GFP-trap assay. Strikingly, in *PACSIN1* KO cells, the amount of endogenous VAMP8 that co-precipitated with GFP-STX17 was decreased, but the amount of co-precipitated endogenous SNAP29 was unaltered (Fig 5A and 5B). Regarding the YKT6-SNAP29-STX7 complex, the amount of endogenous STX7 that was co-precipitated with GFP-YKT6 was also decreased (Fig 5C and 5D). These results together indicates that trans-SNARE assembly between Qa-SNARE and R-SNARE is commonly impaired in *PACSIN1* KO cells. We noticed that YKT6-SNAP29 interaction increased in *PACSIN1* KO cells, although the exact reasons of this is currently unknown. It could be possible that PACSIN1 might be involved in another biological processes through those SNARE components. For instances, it has been shown YKT6 and SNAP29 have been known to be also

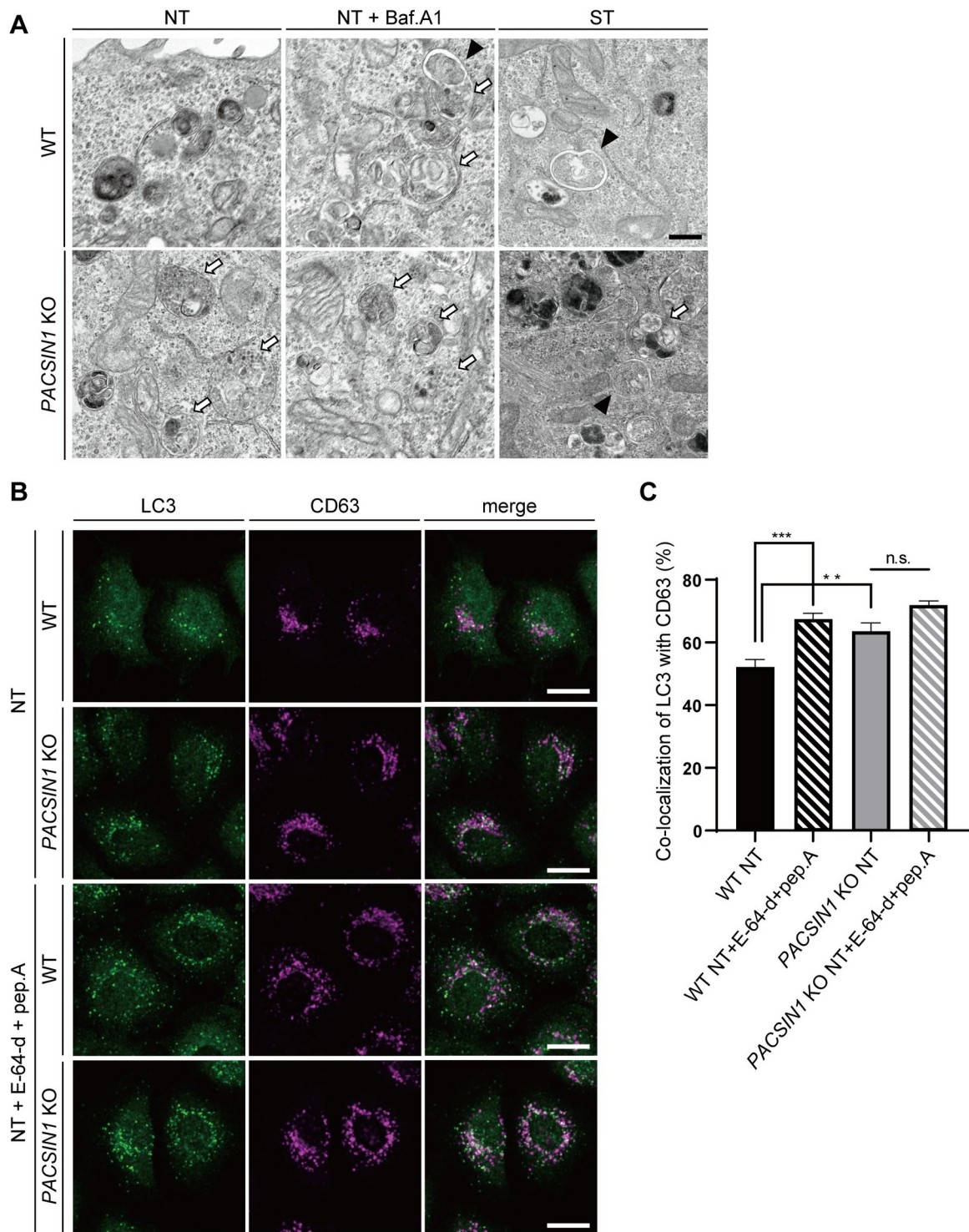

**Fig 3.** ***PACSIN1* depletion causes amphisome accumulation.** (A) WT or *PACSIN1* KO HeLa cells were cultured for 2 h in growth medium or EBSS, or for 6 h with 125 nM Baf.A1 in growth medium. After fixation, cells were analyzed by electron microscopy. Autophagosomes are indicated by arrowheads, and amphisomes are indicated by arrows. Scale bars, 500 nm. (B) WT or *PACSIN1* KO HeLa cells were cultured for 2 h in growth medium with or without 10 μg/mL E-64-d and pepstatin A. After fixation, cells were co-immunostained with anti-LC3 and anti-CD63 antibodies. Scale bars, 20 μm. (C) The co-localization rate of LC3 with CD63 was quantified using CQ1 software, mean ± s.e.m. More than 200 cells were analyzed per condition in each experiment (n = 5). n.s.; not significant, $^{**}p < 0.01$, $^{***}p < 0.001$ (one-way ANOVA with Tukey's multiple comparisons test).

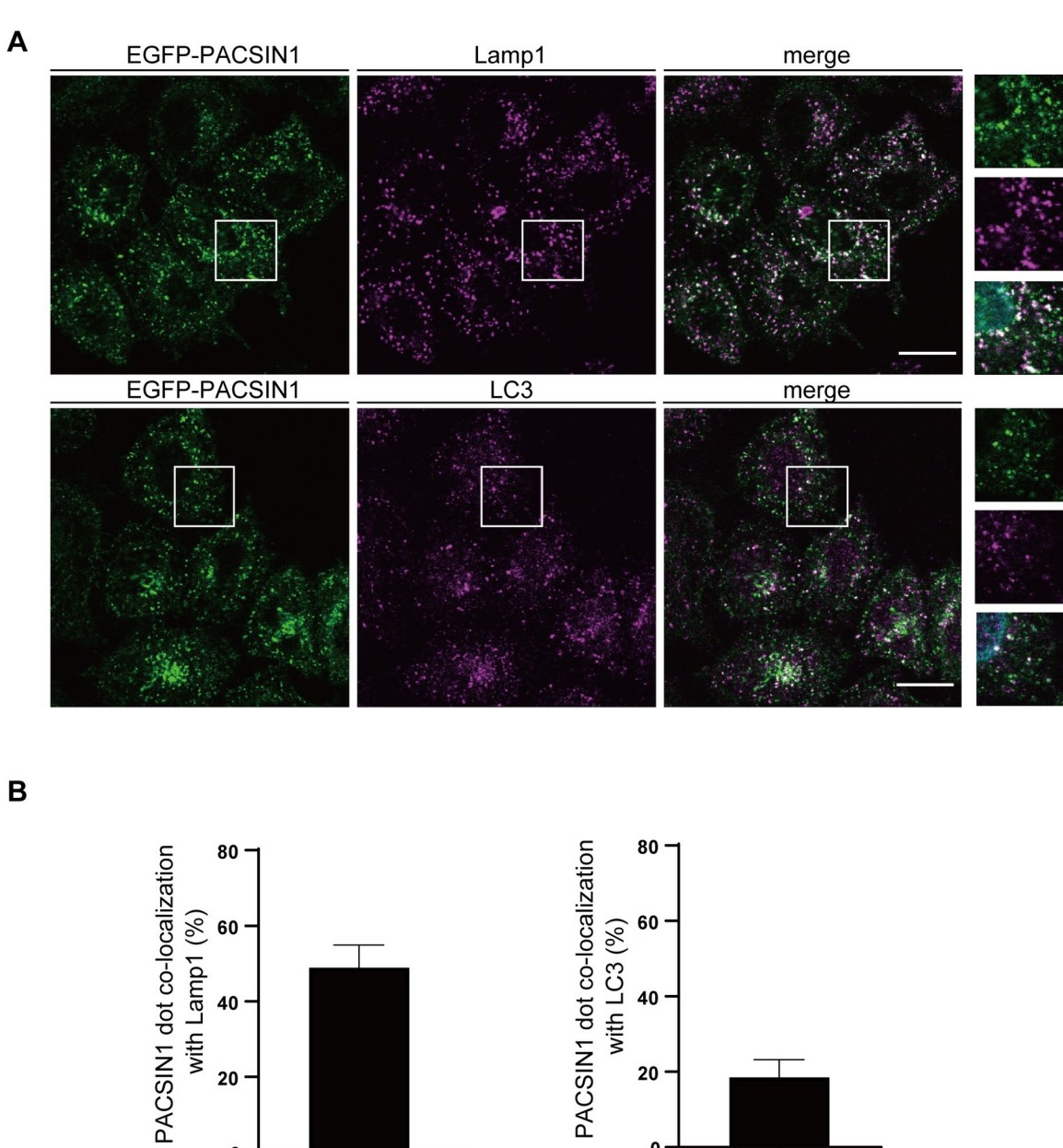

**Fig 4. PACSIN1 localizes to both autophagosomes and lysosomes.** (A) WT HeLa cells stably expressing EGFP-PACSIN1 were cultured for 2 h in growth medium with 125 nM Baf.A1. Cells were pre-treated with 0.05% saponin and fixed. Cells were then stained with anti-LC3 or anti-Lamp1 antibodies, and analyzed by confocal microscopy. Scale bars, 20 μm. (B) The co-localization rate of PACSIN1 with Lamp1 or LC3 was quantified by Fiji, mean ± s.e.m. More than 200 cells were analyzed per condition in each experiment (n = 3).

involved in exosome or autophagosome secretion [30–32] and whether PACSIN1 is involved in such process needs to be clarified in future. Nevertheless, our results suggest that PACSIN1 is required for assembly and/or stabilization of both trans-SNARE complexes. Conceivably, the disassembly of the trans-SNARE complexes leads to defective amphisome-lysosome fusion in *PACSIN1* KO cells.

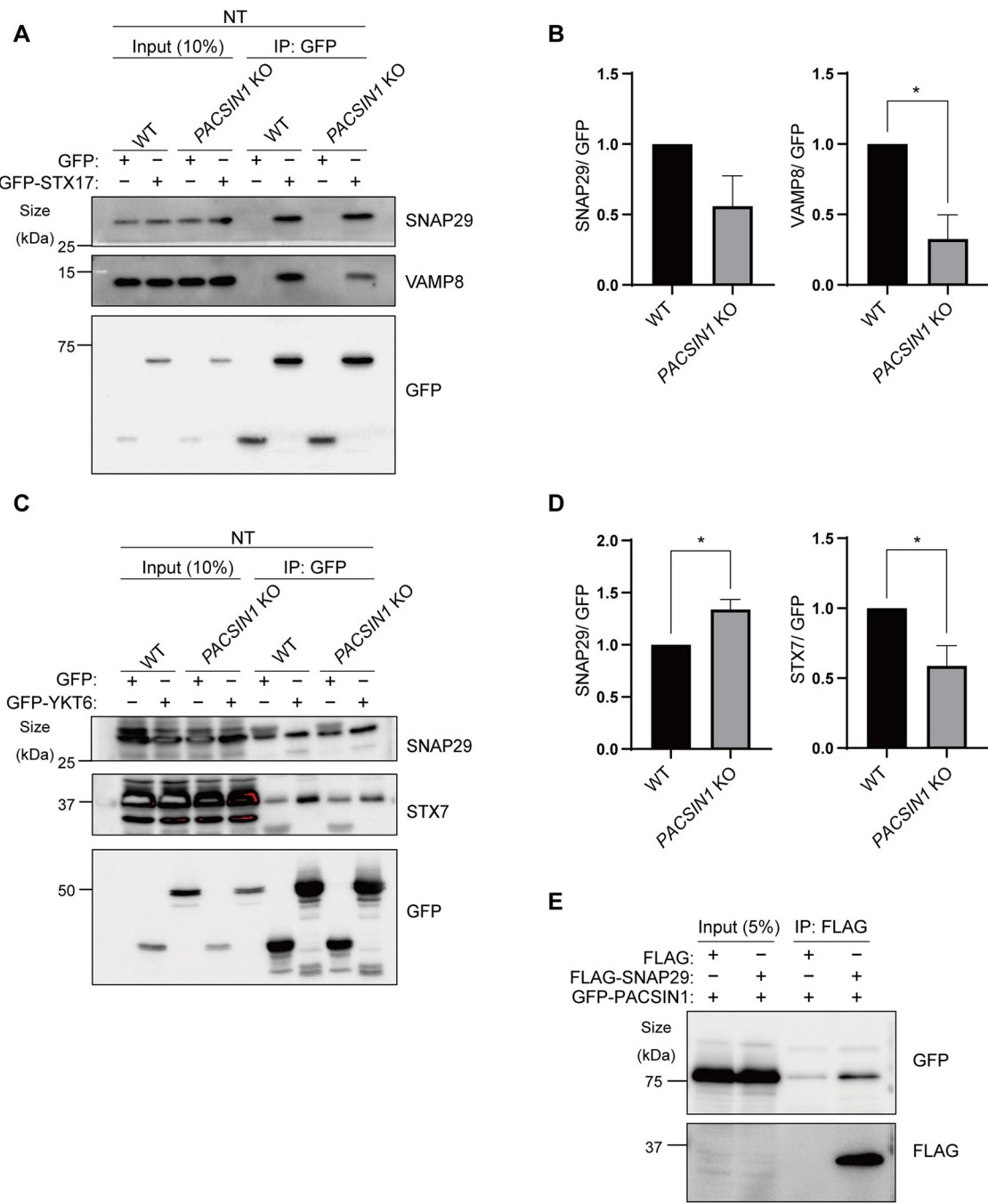

**Fig 5. PACSIN1 is essential for assembly of SNARE complex.** (A) WT or *PACSIN1* KO HeLa cells were transfected with GFP-STX17. The lysates were immunoprecipitated with GFP-trap beads and immunoblotted with the indicated antibodies. (B) Quantification data show the amount of SNAP29 or VAMP8 normalized by GFP. Mean ± s.e.m (n = 3). *$p < 0.05$ (two-tailed, unpaired t-test). (C) WT or *PACSIN1* KO HeLa cells were transfected with GFP-YKT6. The lysates were immunoprecipitated with GFP-trap beads and immunoblotted with the indicated antibodies. (D) Quantification data show the amount of SNAP29 or STX7 normalized by GFP. Mean ± s.e.m (n = 3). *$p < 0.05$ (two-tailed, unpaired t-test). (E) WT HeLa cells were transfected with FLAG-SNAP29 and GFP-PACSIN1. The lysates were immunoprecipitated with FLAG-M2 beads and immunoblotted with anti-GFP antibody.

To further examine the relationship between both SNARE complexes and PACSIN1, we focused on SNAP29, which is the common component of these SNAREs, because the assembly of both SNAREs was decreased in *PACSIN1* KO cells. Importantly, GFP-PACSIN1 was co-precipitated with FLAG-SNAP29 (Fig 5E), while GFP-PACSIN1 did not interact with other SNAREs (S5A Fig). This implies that PACSIN1 contributes to SNARE assembly via interaction with SNAP29.

Some tethering factors, such as PLEKHM1, the HOPS complex, and EPG5, promote assembly or stabilization of SNARE complexes during autophagosome-lysosome fusion [11–15]. Therefore, we also examined whether *PACSIN1* deletion affected the interaction of these tethering factors. The result of a GFP-trap assay using GFP-PLEKHM1 showed that PLEKHM1 interacted with GABARAP, Rab7, and the HOPS complex, as in previous reports [13,33]. This suggests that recruitment of the tethering factors PLEKHM1 and the HOPS complex was unaffected in *PACSIN1* KO cells (S5B Fig) Furthermore, we also checked STX17-SNAP25-VAMP8 interaction, because, depletion of EPG5, which is required for STX17 complex assembly, causes abnormal fusion of autophagosome with endocytic vesicles and promotes assembly of STX17-SNAP25-VAMP8 complex [14]. The result of immunoprecipitation assay using by SNAP25-FLAG exhibited that interaction between SNAP25-FLAG and STX17/VAMP8 in *PACSIN1* KO were comparable to WT cells (S5C Fig). These results suggest that at least the recruitment of several tethering factors was not impaired in *PACSIN1* KO cells.

We also investigated whether PACSIN1 acts as a Rab7 effector to be recruited to lysosomes. We examined the interaction between PACSIN1 and Rab7 using a yeast two-hybrid assay. Contrary to our expectations, PACSIN1 did not interact with Rab7$^{GTP}$ (S5D Fig), suggesting that PACSIN1 is recruited to lysosomes independently of Rab7.

## PACSIN1 regulates lysophagy, aggrephagy but not mitophagy

Our results indicate that PACSIN1 is required for amphisome-lysosome fusion during nutrient-rich basal autophagy but not starvation-induced autophagy, suggesting that the PACSIN1-mediated fusion process depends on external stimuli. To further clarify this phenomenon, we examined the roles of PACSIN1 in other autophagy-inducing conditions. Lysophagy is a type of selective autophagy induced by lysosomal damage caused by many intrinsic and extrinsic materials and is essential for lysosomal homeostasis [34]. In the experimental condition, lysosomal damage is often induced by the lysosomotropic compound L-Leucyl-L-leucine methyl ester (LLOMe), which is known to accumulate in lysosomes and is converted into its membranolytic form by a lysosomal thiol protease, leading to lysosomal damage [34–39]. Upon induction of lysophagy, autophagosomes sequester damaged lysosomes to maintain lysosomal and cellular homeostasis. We used tf-Galectin-3 (Gal3) as a reporter to monitor the clearance of damaged lysosomes [34,40]. Gal3 is a β-galactose–binding lectin that can be used as a marker of damaged endo/lysosomes. When endo/lysosomal membranes rupture and luminal β-galactose is exposed to the cytosol, cytosolic Gal3 is recruited to the ruptured membranes. After induction of lysosomal damage using LLOMe, tf-Gal3 puncta appear on damaged lysosomal membranes, and, similar to tf-LC3, turn red when they are delivered to acidic lysosomes. Ten hours after LLOMe treatment, *PACSIN1* KO cells exhibited delayed attenuation of GFP signals compared with WT cells, behavior similar to that of *ATG13* KO and *FIP200* KO cells, both of which exhibit autophagy depletion (Fig 6A and 6B). This result suggests that PACSIN1 is necessary for lysophagy. Importantly, *PACSIN1* KO cell showed reduced interaction of STX17 with VAMP8 and SNAP29, respectively (S6A Fig). Intriguingly, the interaction of YKT6 with STX7 and SNAP29 were not detected during lysophagy (S6B Fig). These results suggest that STX17 complex rather than YKT6 complex is mainly utilized for lysophagy and PACSIN1 is required for the assembly of STX17 complex.

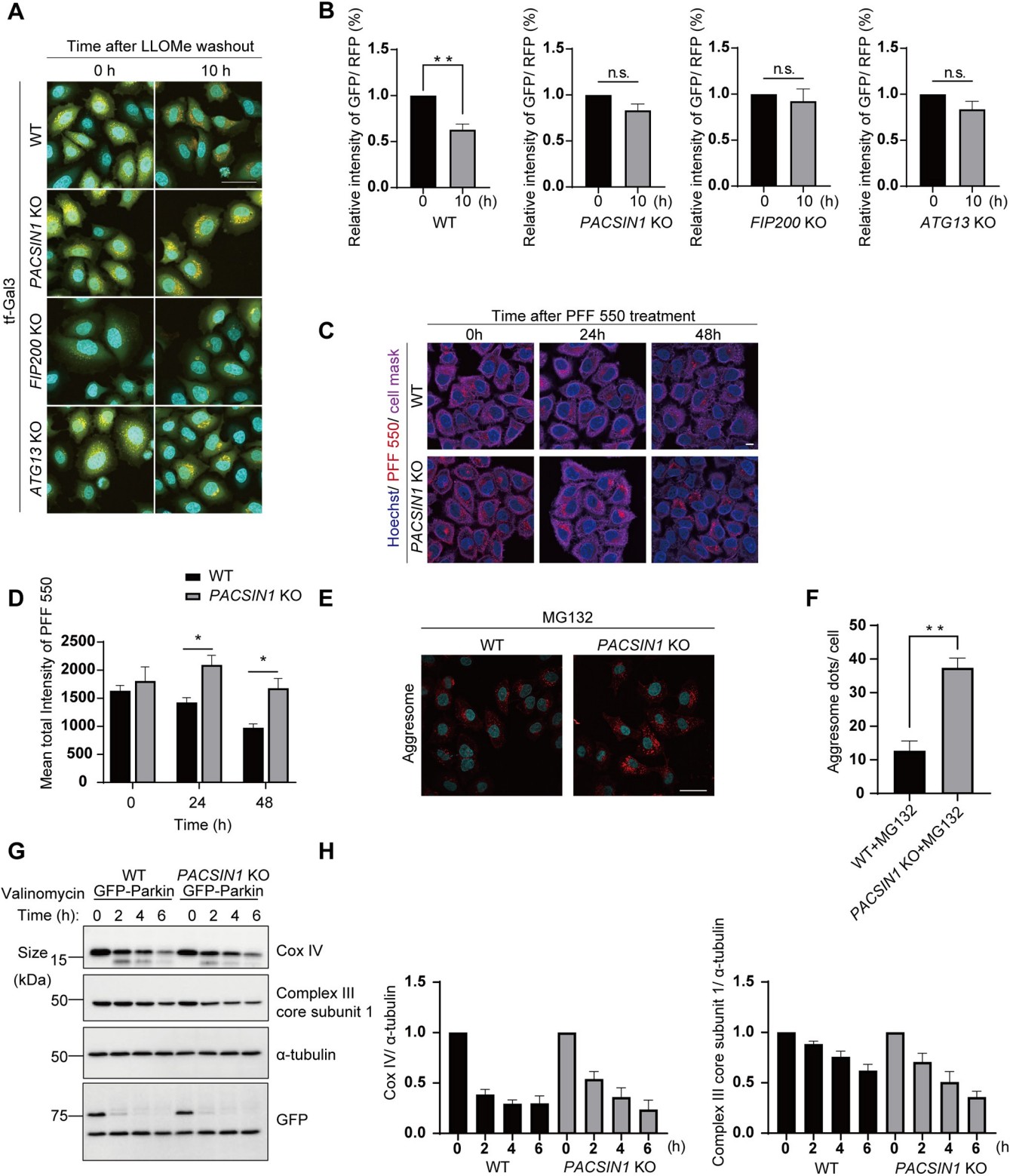

**Fig 6. PACSIN1 is indispensable for lysophagy, aggrepahgy but not mitophagy.** (A) WT, *PACSIN1* KO, *FIP200* KO, and *ATG13* KO HeLa cells stably expressing tf-Gal3 were cultured for 0 h or 10 h in growth medium after 1 mM LLOMe treatment for 1 h. Scale bars, 50 μm. (B) After fixation, cells were analyzed and the relative mean intensity of GFP/RFP was measured using CQ1 software. More than 100 cells were analyzed per condition in each experiment (n = 3), mean ± s.e.m. n.s.; not significant. *P* value (\*\**p* < 0.01) was determined by the two-tailed, unpaired t-test. (C) WT or *PACSIN1* KO HeLa cells were

treated with α-synuclein pre-formed fibril labeled with ATTO 550 (PFF 550) for 1 h, and 0, 24, and 48 h after treatment cells were fixed and stained with indicated dyes. Scale bar, 10 μm. (D) Mean total intensity of PFF 550 in cytosol were measured by Cell Profiler, n > 100 cells from three independent experiments. *P* value (*$p < 0.05$) was determined by two-tailed, unpaired t-test. mean ± s.e.m. (E) Confocal images of WT and *PACSIN1* KO HeLa cells treated with 20 μM MG132 for 6 h. The samples were stained with aggresome detection dye. Scale bars, 40 μm. (F) The number of aggresome dots normalized per cell were quantified by Fiji, mean ± s.e.m.; More than 100 cells were analyzed per condition in each experiment (n = 3). *P* value (**$p < 0.01$) was determined by two-tailed, unpaired t-test. (G) WT or *PACSIN1* KO HeLa cells stably expressing GFP-Parkin were treated with 1 μM valinomycin for 0, 2, 4, or 6 h. After each time period, cells were lysed with lysis buffer and immunoblotted with the indicated antibodies. (H) Quantification data show the amount of Cox IV or Complex III core subunit 1 normalized by α-tubulin. Values represent mean ± s.e.m (n = 3).

We further examined whether PACSIN1 is involved in the degradation of other substrates by autophagy. An aggregation prone protein, α-synuclein which is the main factor for Parkinson disease is known to be degraded by autophagy [41–44]. To analyze role of PACSIN1 on degeneration of α-synuclein aggregates, we analyzed the degradation rate of internalized α-synuclein pre-formed fibril labeled with ATTO 550 (PFF 550) [45]. We treated WT or *PACSIN1* KO HeLa cells with PFF 550 for 1 h, and the remaining internalized PFF 550 were evaluated at 0, 24, 48 h after treatment. We found that the internalized PFF 550 remains more abundantly in *PACSIN1* KO cells, suggesting the delay of autophagic flow against α-synuclein aggregates (Fig 6C and 6D). In addition, we examined the role of PACSIN1 in degradation of misfolded protein induced by inhibition of ubiquitin-proteasome system (MG132) or protein translation (puromycin) [46–48]. We found that *PACSIN1* KO showed increased protein aggregates detected by aggresome detection kit or Western blot for p62 and Ubiquitin, respectively (Figs 6E and 6F and S7). These results collectively indicate that PACSIN1 is also required for aggrephagy.

Furthermore, we also evaluated the possible involvement of PACSIN1 in mitophagy, another type of selective autophagy. GFP-Parkin–expressing HeLa cells were treated with valinomycin, a classical respiratory chain uncoupler and inducer of mitophagy. After valinomycin treatment, the amount of the mitochondrial inner membrane proteins Cox IV and complex III core subunit 1 similarly decreased with time in both WT and *PACSIN1* KO cells (Fig 6G and 6H), suggesting that PACSIN1 was dispensable for Parkin-dependent mitophagy. These results indicate that PACSIN1 functions during specific selective autophagy.

## PACSIN1 is required for basal autophagy and clearance of protein aggregate in *C. elegans*

To confirm the role of PACSIN1 *in vivo*, we examined whether *sdpn-1*, a homolog of PACSINs in *C. elegans*, was required for autophagy. An *sdpn-1* mutant exhibited accumulation of LGG-1, which is used as an autophagosome marker and a homolog of yeast Atg8 in worms (Fig 7A and 7B), suggesting that PACSIN1 contributes to autophagy *in vivo*.

Next, we examined the pathophysiological role of PACSIN1 in worms. Worms expressing α-synuclein::YFP in muscles, exhibited accumulation of α-synuclein aggregates with aging. *sdpn-1* mutant worms that expressed α-synuclein::YFP showed greater accumulation of aggregates than WT worms (Fig 7C and 7D).

In addition, to analyze the decreased motility that worsens with age and is exacerbated by α-synuclein aggregation, we performed a bending assay. *sdpn-1* mutants expressing α-synuclein::YFP in day 1 showed similar motility as WT worms (Fig 7E, S1 and S2 Movies). By contrast, *sdpn-1* mutants in day 5 exhibited a significantly lower bending frequency compared with WT worms (Fig 7F, S3 and S4 Movies), suggesting that the acceleration of age-dependent impairment of locomotion activity in *sdpn-1* mutants was due to defective aggregation clearance.

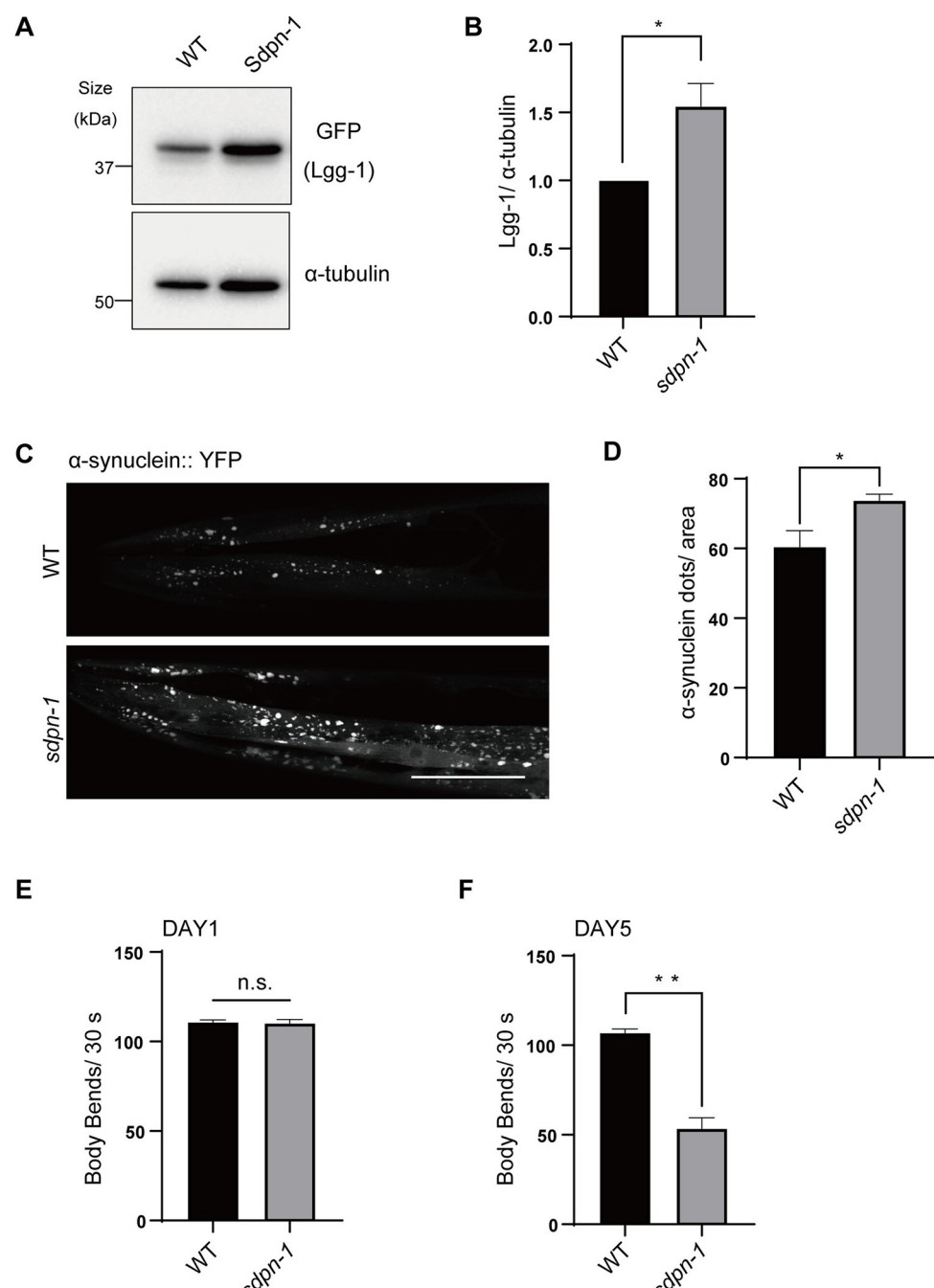

**Fig 7. PACSIN1 is required for basal autophagy and the clearance of aggregation prone proteins in *C. elegans*.** (A) WT and *sdpn-1* mutants expressing GFP::Lgg-1 were used for immunoblot assays at the day 1 adult stage. The lysates were immunoblotted with anti-GFP and anti–α-tubulin antibodies. (B) Quantification data showed the amount of Lgg-1 normalized by α-tubulin. Values represent mean ± s.e.m (n = 3). *P* value (*$p < 0.05$) was determined by the two-tailed, unpaired t-test. (C) WT and *sdpn-1* mutants expressing α-synuclein::YFP were observed at the day 5 adult stage. Scale bars, 50 μm. (D) The number of α-synuclein dots per area was quantified in indicated strains. More than 20 worms were analyzed and the experiments were repeated four times, mean ± s.e.m. *P* value (*$p < 0.05$) was determined by the two-tailed, unpaired t-test. (E) The locomotion of animals (WT and *sdpn-1* mutants expressing α-synuclein:: YFP) at the day 1 adult stage was recorded for 30 s in M9 buffer. The number of body thrashes was counted for more than 20 worms per condition. Values represent mean ± s.e.m. (n = 3). n.s.; not significant (the two-tailed, unpaired t-test). (F) The locomotion of animals (WT and *sdpn-1* mutants expressing α-synuclein::YFP) at the day 5 adult stage was recorded for 30 s in M9 buffer. The number of body thrashes was counted for more than 20 worms per condition. Values represent mean ± s.e.m. (n = 3). *P* value (**$p < 0.01$) was determined by the two-tailed, unpaired t-test.

## Discussion

Autophagosomes mature into autolysosomes by one of two paths: direct fusion with lyso-somes, or fusion with endosomes to form amphisomes, which then fuse with lysosomes [4–6]. However, the functional and molecular differences between these routes and their physiologi-cal relevance are largely unknown, mainly due to the technical difficulty of distinguishing the two paths.

In this study, we demonstrated that PACSIN1 specifically regulated amphisome-lysosome fusion. PACSIN1 was indispensable for the assembly of STX17-SNAP29-VAMP8 and YKT6-SNAP29-STX7 SNARE complexes, which executed autophagosome and endo/lysosome fusion. In addition, PACSIN1 interacted with SNAP29, which is a common component of both SNARE complexes. We found that F-BAR domain of PACSIN1 is required for proper localization (S5E Fig), suggesting that this domain has a critical function. However, since we also observed that PACSINs lacking either F-BAR or SH3 domain, can still interact with SNAP29 (S5F Fig), we could not completely rule out the possibility that small amount of PAC-SIN1 lacking F-BAR localizes on amphisomes/lysosomes and functions to assemble SNARE. Another specific protein-protein interaction module of PACSINs, which NPF motif was con-tained in PACSIN1 and 2 but not 3, may act as the interacting domain with SNAP29. PAC-SIN1 might accelerate the assembly of the STX17 and YKT6 SNARE complexes through interaction with SNAP29. Tethering factors act as a bridge among the vesicles by interacting both with SNARE complexes and the targeted membrane. They also not only stabilize the SNARE complex assembly but also determine the specificity of the fusion membrane [49–51]. Previous reports showed that various tethering factors such as PLEKHM1, EPG5, and the HOPS complex enhance fusion between the autophagosome-endo/lysosome membrane [11–15]. Hence, we speculated that PACSIN1 promotes SNARE assembly by working together with these tethering factors. However, our results showed that PACSIN1 was dispensable for recruiting these proteins (S5B Fig). Thus, another possibility is that like these proteins, PAC-SIN1 itself functions as a tethering factor, and targets specific membranes. We examined whether PACSIN1 acts as Rab7 effector to localize to lysosomes, but PACSIN1 did not interact with Rab7$^{GTP}$ (S5D Fig). Further analyses are necessary to identify the factors that recruit PACSIN1 to both amphisomes and lysosomes.

Our studies on the roles of PACSIN1 revealed that which of the two autophagic pathways was active depended on the context. Our results indicated that in basal autophagy, character-ized by relatively low autophagic activity, the fusion path via amphisomes is preferentially selected over the direct fusion path. In starvation-induced autophagy, on the other hand, in which autophagic activity is dramatically increased, the direct fusion path might be preferen-tially selected, although another route via amphisomes is not completely shut off. Consistent with this idea, in both basal and starvation condition, the interaction of PACSIN1 with SNAP29 was not altered (S5G Fig). This implied that autophagic activity contributed to decide the preferential usage of the two paths but not completely dependent to one route. In selective autophagy, PACSIN1 was required for substrate degradation in lysophagy and aggrephagy, but not in mitophagy. Although it remains unclear what signal determines the autophagy path selection, we speculate that this selection depends not only on environmental stimuli but also on the cargo involved. Another possibility is that lysosomal activity might affect path selection. Consistent with this idea, we found that lysosomal pH was lower during starvation than in the nutrient-rich condition (Fig 2A). This could partly be due to conformational and functional changes in V-ATPase in response to various environmental conditions [52, 53]. During lyso-phagy, on the other hand, lysosomal acidity is transiently compromised upon lysosomal rup-ture [54]. Thus, different fusion paths might be selected depending on lysosomal activity, and

the PACSIN1-mediated fusion process is especially relevant under reduced lysosomal function. Additionally, the difference in the role of the two paths is currently unknown. One theory is that the path via amphisomes results in slower degradation of substrate than the path involving direct autophagosome fusion with lysosomes. Moreover, because the path via amphisomes could change the route from degradation to secretion, the substrates may function as a secretion signal. Alternatively, amphisomes may act as a storage compartment to temporarily separate the substrate from degradation. Therefore, the availability of different routes may help regulate substrate degradation both temporally and spatially.

Evidence from recent studies suggests that autophagy degrades unnecessary proteins and removes damaged organelles to prevent various human pathologies, including neuronal degeneration [1]. We have shown that a *sdpn-1* mutant exacerbated accumulation of α-synuclein aggregation in *C. elegans*. However, it is unclear if this was a result of decreased basal autophagic activity or a defect in lysophagy and/or aggrephagy. According to previous reports, aggregated proteins damage lysosomal membranes [55]; therefore, the clearance of aggregated proteins depends not only on basal autophagy, but also on lysophagy. In either case, PACSIN1 plays an important role in eliminating aggregated proteins in neuronal cells. In addition, previous studies showed that HTT with expanded polyglutamine repeats (mHTT) sequesters PACSIN1 and disrupts its function, leading to synaptic failure and neuronal loss [56,57]. Basal autophagy is known to decrease with age, causing insufficient digestion of aggregation-prone proteins relevant to neurodegenerative disease. It is worth investigating if such proteins sequester PACSIN1 in vivo and reduce its function, thereby potentially impairing basal autophagy and contributing to neurodegenerative disease.

## Materials and methods

### Reagents and antibodies

The following reagents were purchased from the indicated companies; Baf.A1 (wako, 029–11643), E-64-d (Peptide Institute, 4321-v), and pepstatin A (Peptide Institute, 4397-v), MG132 (Miilipore, 474790), puromycin (Invitrogen, ant-pr-1).

For immunoblotting, the following antibodies were used at the indicated dilutions: anti-LC3 (rabbit, 1/1500; MBL, pM036), anti-p62 (rabbit, 1/5000; MBL, pM045), anti–α-tubulin (mouse, 1/50000; Sigma-Aldrich, T5168), anti-ATG13 (rabbit, 1/1000; Sigma-Aldrich, SAB4200100), anti-SNAP29 (rabbit, 1/1000; Abcam, 138500 (EPR9199)), anti-VAMP8 (rabbit, 1/10000; Abcam, 76021), anti-STX7 (rabbit, 1/1000; Bethyl, A304-512A-T), anti-GFP (rabbit, 1/1000; MBL, 598), anti-UQCRC1 (Complex III core subunit 1) (mouse, 1/2000; Invitrogen, 459140), anti-COXIV (rabbit, 1/1000; CST, 4850 (3E11)), anti-GFP (mouse, 1/2000; Santa Cruz Biotechnology, 9996 (B-2)), anti-PACSINs (rabbit, 1/1000; [23]), anti-PACSIN3 (mouse, 1/100; Santa Cruz Biotechnology, 373952 (F-8)), anti-Rab7 (mouse, 1/1000; CST, 95746s (E907E)), anti-GABARAP (rabbit, 1/1000; MBL, pM037), anti-Vps41 (mouse, 1/100; Santa Cruz Biotechnology, 377118 (D-12)), anti-Vps39 (mouse, 1/100; Santa Cruz Biotechnology, 514762 (C-5)), anti-STX17 (rabbit, 1/1000; Sigma-Aldrich, HPA001204), anti-YKT6p (mouse, 1/100; Santa Cruz Biotechnology, 365732 (E-2)), anti-FK2 (mouse, 1/1000; Nippon Bio-test Laboratories, 0918–2), anti-Lamin B (goat, 1/1000; Santa Cruz Biotechnology, 6217(M-20)), and anti-FLAG (rabbit, 1/2000; Sigma-Aldrich, F7425).

The secondary antibodies used for immunoblotting were as follows: HRP-conjugated goat anti–rabbit IgG (1/5000; Jackson ImmunoResearch, 111-035-003) and HRP-conjugated goat anti–mouse IgG (1/5000; Jackson ImmunoResearch, 115-035-003).

For immunofluorescence staining the following antibodies were used at the indicated dilutions: anti-LC3 (rabbit, 1/1000; MBL, pM036), anti-Lamp1 (mouse, 1/1000; Santa Cruz

Biotechnology, 20011 (H4A3)), and anti-CD63 (mouse, 1/500; BD Biosciences, 556019). The secondary antibodies used for immunofluorescence staining were as follows: goat anti–mouse IgG H&L Alexa Fluor488 (1/2000; Abcam, 150117), goat anti–rabbit IgG H&L Alexa Fluor488 (1/2000; Abcam, 150085), and goat anti–rabbit IgG H&L Alexa Fluor568 (1/2000; Abcam, 175695).

For immuno-EM staining following antibodies were used with indicated dilution in this study: anti-GFP antibody (rat, 1/200; nacalai tesuque, 04404–84), Nanogold-Fab' goat anti rat IgG(H+L) (1/400; Nanoprobes, 2008).

## Plasmids and retroviral infections

The pMRx-IRES-puro and pMRx-IRES-bsr vectors were kindly provided by S. Yamaoka (Tokyo Medical and Dental University, Tokyo, Japan). pMRX constructs were generated to encode tandem fluorescent-tagged LC3 (tf-LC3) [58], tandem fluorescent-tagged Gal3 (tf-Gal3) [34], and Parkin (Addgene, #23955), Lamp1 [58] and LC3B [35]. Recombinant retroviruses were prepared as previously described [59]. Stable transformants were selected in growth medium with 1.5 μg/mL puromycin or 3 μg/mL blasticidin. For co-immunoprecipitation experiments, GFP-tagged STX17 [60] and YKT6, and 3xFLAG-tagged SNAP29 were subcloned into pcDNA3.1. pcDNA3.1 was purchased from Invitrogen (V79020). Human PACSIN1 (Addgene, #20545) was subcloned into pMRX, pcDNA3.1, and pACT2 (Clontech, 638822). PACSIN1 mutants, ΔF-BAR (1–277) and ΔSH3 (393–444) were subcloned into pcDNA3.1. pEGFP-C1-PLEKHM1 was obtained as previously described [33]. The DNA fragments of *SNAP29*, *YKT6* and *SNAP25A* were amplified with the following specific primers using U2OS and HeLa cells, respectively:

*SNAP29*_Fw; 5′- ctttgaattcggatccatgtcagcttaccctaaaagc-3′
*SNAP29*_Rv; 5′- agaaagctgggtcgactcagagttgtcgaacttttctttc-3′
*YKT6*_Fw; 5′- ctttgaattcggatccatgaagctgtacagcctcag-3′
*YKT6*_Rv; 5′- agaaagctgggtcgactcacatgatggcacagcatg-3′
*SNAP25A*_Fw; 5′- attcggatccctcgaggctaccatggccgaagac-3′
*SNAP25A*_Rv; 5′- gtaccgcatgcggccgcaccacttcccagcatctttg-3′

## Cell culture and transfection

HeLa Kyoto and plat-E cells were cultured in Dulbecco's modified Eagle's medium, DMEM (Sigma-Aldrich, D6429) supplemented with 10% fetal bovine serum, 1% L-glutamine, and 50 μg/mL penicillin-streptomycin in a 37°C, 5% CO2 incubator. The Plat-E cells were provided by T. Kitamura (The University of Tokyo, Japan). HeLa Kyoto cells were provided by S. Narumiya (Kyoto University, Japan). Transient transfections were carried out using Lipofectamine 2000 (Invitrogen), and cells were used in experiments 24 h after transfection. For nutrient starvation, cells were cultured in EBSS (Sigma-Aldrich, E2888) for 2 h.

## Generation of KO cell lines by CRISPR-Cas-9

*PACSINs* KO HeLa cell lines were generated using the following guide RNAs (gRNAs):

*PACSIN1*, 5′-TCACCGACTGGGCCAAGCGT-3′
*PACSIN2*, 5′-GTCACATATGATGATTCCGT-3′
*PACSIN3*, 5′-ACTACAGGCGCACGGTACAG-3′.

Annealed gRNA oligonucleotides were inserted into px458 vectors and the plasmids were transfected into HeLa cells using ViaFect (Promega). KO cells were verified by genomic DNA sequencing. *PACSIN1* KO clone #1 was used in most of the experiments, unless otherwise stated.

## Immunoblotting

HeLa cells were washed with ice-cold PBS twice and lysed with RIPA Buffer (50 mM Tris-HCl (pH 8.0), 150 mM NaCl, 1% Triton X-100, 0.1% SDS, 0.5% sodium deoxycholate) containing 1x protease inhibitor cocktail (Roche) and 1 mM phenylmethylsulfonyl fluoride. The samples were adjusted to equal concentrations and then subjected to SDS-PAGE and transferred to polyvinylidene difluoride membranes. The membranes were blocked with TBS-T containing 1% skim milk for 30 min and incubated overnight at 4˚C with primary antibodies diluted in blocking solution. The membranes were washed three times with TBS-T and then incubated for 1 h at room temperature with HRP-conjugated secondary antibodies in blocking solution. After washing the membranes three times with TBS-T, immunoreactive bands were detected using Luminate Forte (Merck Millipore, WBLUF0100) or ImmunoStar LD (FUJIFILM, 290–69904) by a ChemiDoc Touch imaging system (Bio-Rad). The autophagic flux was calculated by subtracting the LC3-II value, which normalized by α-tubulin, in the absence of Baf.A1 from that in the presence of Baf.A1 in basal or starvation medium both in WT and *PACSIN1* KO cells. Finally, the relative value normalized to WT in basal condition was calculated.

## Immunofluorescence analysis

HeLa cells were fixed with 4% PFA for 20 min at room temperature. For PACSIN1 localization analysis, WT HeLa cells stably expressing EGFP-PACSIN1 were cultured for 2 h in growth medium with 125 nM Baf.A1 and then pre-permeabilized with 0.05% saponin in PIPES buffer for 5 min before cells were fixed. Fixed cells were permeabilized with 50 μg/mL digitonin-PBS for 5 min, blocked with 0.2% gelatin-PBS for 30 min, and incubated with primary antibodies in 0.2% gelatin-PBS for 1 h at room temperature. Cells were washed with PBS and incubated with fluorescence-conjugated secondary antibodies in 0.2% gelatin-PBS for 1 h at room temperature. The samples were mounted using VECTASHIELD (Vector Laboratories), and observed using an FV3000 confocal microscope (Olympus). The LC3 dot count normalized per cell and co-localization rate of PACSIN1 with organelle markers were analyzed by Fiji (version 2.0.0-rc-69/1.52p).

To analyze co-localization of LC3 with CD63 or Lamp1, cells stably expressing Lamp1-mcherry were cultured in 96-well glass-bottom plates (PerkinElmer) for 2 h in growth medium with or without 10 μg/mL E-64-d and pepstatin A. After fixation, cells were immunostained with anti-LC3 and anti-CD63 antibodies. CQ1 software version 1.05.01.01 (Yokogawa) was used to observe the samples and quantify the co-localization rate of LC3 with Lamp1 or CD63.

## Proteinase K protection assay

HeLa cells were cultured with growth medium or EBSS with or without 250 nM Baf.A1 for 8 h. Cells were collected and resuspended in buffer (20 mM HEPES-KOH (pH 7.4), 220 mM mannitol, 70 mM sucrose, 1 mM EDTA-KOH), then homogenized using a syringe with a 27-gauge needle (10 strokes). Cell lysates were centrifuged at 500 G for 5 min at 4˚C. The collected supernatant was treated with 25 μg/mL proteinase K with or without 0.2% Triton X-100 for 10 min on ice. The samples were precipitated with 10% trichloroacetic acid, then the pellet was washed twice with cold acetone, resuspended in sample buffer, and analyzed by immunoblotting.

## Co-immunoprecipitation assay

After 24 h transfection, cells transiently expressing the indicated tag-protein were washed with PBS twice and then lysed with lysis buffer (10 mM Tris-HCl (pH 7.5), 150 mM NaCl, 0.5 mM EDTA, 0.5% NP-40) containing 1x protease inhibitor cocktail (Roche) and 1 mM phenylmethylsulfonyl fluoride. After 30 min incubation on ice, cell lysates were centrifuged at 20,000

G for 10 min at 4˚C. Collected supernatants were immunoprecipitated by GFP-trap agarose beads (Chromotek, gta-20) for 1 h at 4˚C. After incubation, beads were washed with wash buffer (10 mM Tris-HCl (pH 7.5), 150 mM NaCl, 0.5 mM EDTA) five times and then analyzed by immunoblotting.

### Lysotracker and Magic Red cathepsin B assay

Cells were cultured in 96-well glass-bottom plates (PerkinElmer), treated with 50 nM Lysotracker Red DND-99 (Molecular Probes, L7528) in growth medium with or without 125 nM Baf.A1 or EBSS for 2 h. Cells were fixed with 4% PFA for 20 min and immunostained with anti-Lamp1 antibody. CQ1 software (Yokogawa) was used to observe the cells and calculate the mean intensities of Lysotracker on Lamp1 per cell. For the Magic Red cathepsin B assay, cells were incubated in growth medium with or without 125 nM Baf.A1 or EBSS for 1.5 h and then Magic Red (Immunochemistry Technologies) was added to the medium. After 30 min additional incubation, cells were fixed with 4% PFA for 20 min, and stained with DAPI. CQ1 software (Yokogawa) was used to observe the cells and calculate the mean intensities of Magic Red dots per cell.

### Electron microscopy

Cells were cultured for 2 h in growth medium or EBSS, or with 125 nM Baf.A1 for 6 h, and fixed with 2.5% glutaraldehyde (Wako, 071–01931) in PBS and then in 2% OsO4 solution. After the samples were embedded in Quetol812 (Nisshin EM), 80-nm ultrathin sections were obtained using an Ultracut E ultramicrotome (Reichert-Jung). These sections were stained with a solution of uranyl acetate and lead, and observed using a transmission electron microscope (Hitachi, H-7650).

For immuno-electron microscopic analysis, *PACSIN1* KO cells stably expressing EGFP-LC3B were cultured on a Cell Desk LF1(Sumitomo Bakelite). Cells were fixed with 4% paraformaldehyde in 0.1 M sodium-phosphate buffer, pH 7.4 (PB) for 1 h and washed for 5 min three times with 4% Sucrose in 0.1 M PB. Fixed cells were permeabilized in 0.1 M PB containing 0.25% (w/v) saponin for 30 min, and blocked in 0.1 M PB containing 0.005% (w/v) saponin, 10% (w/v) BSA, and 10% (w/v) serum for 10 min. After that, cells were stained with anti-GFP antibody in blocking solution for 18 h at 4˚C, and washed for 10 min six times in 0.1 M PB containing 0.005% (w/v) saponin. Then, cells were stained for 2 h at room temperature with an anti-rat IgG conjugated to 1.4 nm gold particle in blocking solution, and washed for 10 min six times in 0.1 M PB containing 0.005% (w/v) saponin. Cells were fixed for 10 min with 1% (w/v) glutaraldehyde in 0.1 M PB and washed three times for 5 min each with PBS containing 50 mM glycine, PBS containing 1% (w/v) BSA and distilled water. Cells were treated with the gold enhancement mixture using GoldEnhance EM kit (Nanoprobes) for 5min at room temperature to increase the size of gold particles and improve visualization by electron microscopy, and washed with distilled water. Cells were then post-fixed for 1 h with 1% (w/v) OsO$_4$ and 1.5% (w/v) potassium ferrocyanide in 0.1 M PB, and then dehydrated in a graded series of ethanol and embedded in epoxy resin (TAAB Laboratories Equipment). Ultra-thin (85 nm) sections of cells were stained with 2% (w/v) uranyl acetate for 1 h, and then stained with lead citrate solution for 2 min. Electron micrographs were obtained with transmission electron microscope (Hitachi, H-7650).

### tf-LC3 and tf-Gal3 assays

Cells stably expressing tf-LC3 were cultured in 96-well glass-bottom plates (PerkinElmer), treated with growth medium with or without 250 nM Baf.A1 for 2 h, fixed with 4% PFA for 20 min, and then observed using CQ1 software (Yokogawa).

For lysophagy assays, cells stably expressing tf-Gal3 were cultured in 96-well glass-bottom plates, and treated with 1 mM LLOMe (Sigma-Aldrich, L7393) for 1 h. Then, LLOMe was washed out with DMEM and cells were incubated in DMEM without LLOMe for 0 or 10 h. After incubation for the indicated times, cells were fixed with 4% PFA for 20 min. CQ1 software (Yokogawa) was used to observe the cells and calculate the numbers of tf-LC3 and the mean intensity of tf-Gal3 dots per cell.

## Mitophagy assay

Cells stably expressing EGFP-Parkin were treated with 1 μM valinomycin (Sigma-Aldrich, V0627) for 0, 2, 4, or 6 h. After each time period, cells were lysed with lysis buffer.

## *C. elegans* growth conditions and strains

Nematodes were cultured at 20˚C on nematode growth medium (NGM) agar plates with *E.coli* strain OP50. The worm strains used in this study were as follows: *DA2123*, *adIs2122(lgg-1p:: GFP::lgg-1; rol-6(su1006))*, *HZ589*, *bpIs151(T12G3.1::gfdcp;unc-76) IV; him-5(e1490) V*, *NL5901*, *pkIs2386 [unc-54p:: α-synuclein::YFP + unc-119(+)]* and those strains crossed with *RB1460*, *sdpn-1(ok1667)*, which was a kind gift from Prof. BD Grant (Rutgers University).

## Immunoblotting assays, microscopic observation, and bending assays in worms

Synchronized eggs were obtained from 1–2 h egg lays on NGM plates. For immunoblotting assays, adult animals at the day 1 (GFP::Lgg-1) were lysed with sample buffer, frozen, and thawed, and then the buffer was sonicated. For observation of α-synuclein aggregation, α-synuclein::YFP animals at the day 5 adult stage were anesthetized in 0.5% sodium azide and observed using a FV3000 confocal microscope (Olympus). YFP puncta in predetermined pharyngeal regions were counted using Fiji. In each experiment, images from more than 20 worms per condition were captured. For bending assays, locomotion of animals was recorded for 30 s in M9 buffer. The number of body thrashes was counted for more than 20 worms per condition.

## Yeast two-hybrid assay

A yeast two-hybrid assay was performed as described previously [61]. The *S. cerevisiae* strain PJ69-4A was co-transformed with Gal4BD plasmids (pFBT9-Rab7 GTP-restricted mutant) and Gal4AD plasmids (pACT2) harboring PACSIN1. The cells were grown on SC-LW (SC/ −Leu/−Trp) plates for 3 days at 30˚C, then five independent colonies were picked and restreaked on SC-LW and QDO (SC/−Leu/−Trp/−His/−Ade) plates, and the cells were grown for 3 days at 30˚C.

## Generation of α-synuclein pre-formed fibril labeled red-fluorescent dye (PFF 550)

Fibrillar α-synuclein were generated as described previously [45] with some modifications. Briefly, recombinant α-synuclein were dissolved in PBS at 5 mg/ml, and incubated with orbital-shaking at 800 rpm at 37˚C for 5 days followed by centrifugation at 100,000 G at 20˚C for 1 h. For fluorescent labelling, fibrillar α-synuclein were labeled with ATTO550 (Echelon) dye according to manufactures protocol before centrifugation. The pellets were resuspended in PBS, measured concentration, aliquoted and stored -80˚C.

### Fibrillar α-synuclein degradation assay

WT or *PACSIN1* KO HeLa cells were seeded in glass-bottom dishes. 24 h after seeding, the cells were incubated with PFF 550 for 1 h, and washed 3 times with fresh cell culture medium to remove remaining fibrils in cell culture medium, and fixed at 0, 24, 48 h after treatment. The cells were stained with Hoechst 34580 for nuclei and Cell Mask for plasma membrane (Invitrogen), and subjected to microscope observation. Intracellular fibrillar α-synuclein were quantified automatically by Cell Profiler.

### Protein aggregates clearance assay

WT or *PACSIN1* KO HeLa cells were treated with 20 μM MG132 or 2.5 μg/mL puromycin for 6 h. Detergent soluble-insoluble proteins were fractionated using 1% triton-X 100 with PBS [62]. After centrifugation, the supernatant was used as the soluble fraction. The pellet was extracted with RIPA buffer included 7 M Urea and used as the insoluble fraction. Aggresome staining was performed by aggresome detection kit (abcam, ab139486) according to the manufacturer's instruction. The number of aggresome dots normalized per cell were quantified by Fiji.

### Cell proliferation assay

Cell proliferation was determined using Cell counting kit-8 (Dojindo Molecular Technologies, 343–076239). 5,000 cells were seeded in 96 well plate. After 24 h, cells were treated with CCK-8 for 2 h according to manufacturer's instruction, and absorbance was measured at 450 nm using a microplate reader, Infinite 200 PRO (TECAN).

### siRNA

The siRNA for PICK1 was purchased from Sigma. The siRNA duplex oligomers for Luciferase and ATG13 were designed as follows as respectively: Luciferase: 5′-UCGAAGUAUUCCGC-GUACG-3′, ATG13: 5′-GAGUUUGGAUAUACCCUUU-3′. The siRNAs (final concentration of 20–50 nM) were transfected into HeLa cells using Lipofectamine RNAiMAX (Invitrogen). 48 h after transfection, the cells were used for western blotting and quantitative PCR.

### RNA extraction and quantitative PCR

Total RNA was extracted using the RNeasy Mini Kit (Qiagen) and cDNA was generated using the iScript cDNA synthesis kit (BioRad). Quantitative PCR was performed using Power SYBR Green (Applied Biosystems) on QuantStudio7 (Applied Byosystems). Primers sequences were as follows:

PICK1_Fw; 5′-GAAGTTCGGCATTCGGCTTC-3′
PICK1_Rv; 5′-CTTGATGGTGAGGCGAGTGT-3′
GAPDH_Fw; 5′-TGCACCACCAACTGCTTAGC-3′
GAPDH_Rv; 5′-GGCATGGACTGTGGTCATGAG-3′

### Statistical analysis

All quantitative data are presented as mean ± s.e.m. Statistical analyses were performed using GraphPad Prism 8.0.

## Supporting information

**S1 Fig. Deletion of *PACSIN1* but not *PACSIN2* or *PACSIN3* is impaired autophagic activity.** (A) Isolated *PACSIN1* KO clone #1 HeLa cells exhibited a 14-base deletion and isolated *PACSIN1* KO clone #2 HeLa cells exhibited hemizygous deletion at the indicated locus on the second exon of *PACSIN1*. The PAM sequence and recognition sequence are labeled in green and red, respectively. (B) Immunoblot confirming the absence of protein in established *PACSIN2* and *PACSIN3* KO HeLa cells. (C) WT or *PACSIN1* KO clone #2 HeLa cells were cultured for 2 h in growth medium (DMEM, NT) or starvation medium (EBSS, ST) with 125 nM Baf. A1 and then analyzed by immunoblot using anti-LC3 and anti–α-tubulin antibodies. *PACSIN1* KO clone #2 HeLa cells exhibited impairment of autophagic activity similar with *PACSIN1* KO clone #1 HeLa cells. (D) Cell proliferation assay performed by WST-8 after 24 h seeded WT and *PACSIN1* KO clone #1, #2 HeLa cells, mean ± s.e.m (n = 3). n.s.; not significant (one-way ANOVA with Dunnett's multiple comparisons test). (E) WT, *PACSIN2*, and *PACSIN3* KO HeLa cells were cultured for 2 h in growth medium with 125 nM Baf.A1 and then analyzed by immunoblotting using anti-LC3 and anti–α-tubulin antibodies. (F) LC3 flux was quantified, mean ± s.e.m (n = 3). n.s.; not significant (one-way ANOVA with Dunnett's multiple comparisons test).
(TIF)

**S2 Fig. PICK1 depletion doesn't affect autophagic activity.** (A) WT or *PACSIN1* KO HeLa cells treated with siLuciferase or siPICK1 were cultured for 2 h in growth medium with or without 125 nM Baf.A1, then analyzed by immunoblot using anti-LC3 and anti–α-tubulin antibodies. (B) LC3 flux was quantified, mean ± s.e.m (n = 4). n.s.; not significant (one-way ANOVA with Dunnett's multiple comparisons test). (C) Gene expression of PICK1 was quantified by real-time PCR (qRT-PCR), mean ± s.e.m (n = 4). $P$ value (****$p < 0.0001$) was determined by one-way ANOVA with Dunnett's multiple comparisons test.
(TIF)

**S3 Fig. *PACSIN1* deletion doesn't affect lysosomal function.** (A) WT or *PACSIN1* KO clone #2 HeLa cells were cultured in growth medium and treated with Magic Red. After fixation, cells were analyzed using CQ1 software. The cell exhibited relatively high and low intensity of Magic Red were indicated by arrows and arrowheads, respectively. Scale bars, 40 μm. (B) Quantified Magic Red mean intensity normalized per cell, mean ± s.e.m. More than 200 cells were analyzed per condition in each experiment (n = 3). n.s.; not significant (the two-tailed, unpaired t-test).
(TIF)

**S4 Fig. *PACSIN1* KO cells show accumulation of amphisome structures.** (A) Quantified the number of amphisome and lysosome in WT and *PACSIN1* KO HeLa cells from total 80 images of two independent experiments. (B) Immunogold particles identifying LC3B are localized in vacuoles containing small vesicles in *PACSIN1* KO HeLa cells. Scale bars, 500 nm. (C) WT or *PACSIN1* KO HeLa cells stably expressing Lamp1-mcherry were cultured for 2 h in growth medium with or without 10 μg/mL E-64-d and pepstatin A. After fixation, cells were immunostained with anti-LC3 antibodies. Scale bars, 20 μm. (D) The co-localization rate of LC3 with Lamp1 was quantified using CQ1 software, mean ± s.e.m. More than 200 cells were analyzed per condition in each experiment (n = 5). n.s.; not significant, *$p < 0.05$, ***$p < 0.001$ (one-way ANOVA with Tukey's multiple comparisons test).
(TIF)

**S5 Fig. PACSIN1 does not interact with autophagic SNAREs other than SNAP29.** (A) WT HeLa cells were transfected with GFP-PACSIN1. The lysates were immunoprecipitated with GFP-trap beads and immunoblotted with the indicated antibodies. (B) WT and *PACSIN1* KO HeLa cells were transfected with GFP-PLEKHM1. The lysates were immunoprecipitated with GFP-trap beads and immunoblotted with the indicated antibodies. (C) WT or *PACSIN1* KO HeLa cells transiently expressing SNAP25-FLAG were cultured in growth medium or starvation medium (EBSS, ST) for 2 h. The lysates were immunoprecipitated with FLAG-M2 beads and immunoblotted with anti-VAMP8 and anti-STX17 antibody. (D) A Y2H assay showed that PACSIN1 did not interact with Rab7$^{GTP}$. (E) WT HeLa cells stably expressing EGFP-PACSIN1 or EGFP-PACSIN1 ΔF-BAR. Cells were cultured for 2 h in growth medium with 125 nM Baf.A1. Cells were pre-treated with 0.05% saponin and fixed. The samples were analyzed by confocal microscopy. Scale bars, 20 μm. (F) WT HeLa cells were transfected with indicated plasmids. The lysates were immunoprecipitated with FLAG-M2 beads and immunoblotted with the indicated antibodies. (G) WT HeLa cells transiently expressing FLAG-SNAP29 and GFP-PACSIN1 were cultured in growth medium or starvation medium (EBSS, ST) for 2 h. The lysates were immunoprecipitated with FLAG-M2 beads and immunoblotted with anti-GFP and anti-FLAG antibody.
(TIF)

**S6 Fig. STX17 complex rather than YKT6 complex is required for lysophagy.** (A) WT or *PACSIN1* KO HeLa cells transiently expressing GFP-STX17 were treated with 1 mM LLOMe for 1 h. The lysates were immunoprecipitated with GFP-trap beads and immunoblotted with the indicated antibodies. (B) WT or *PACSIN1* KO HeLa cells transiently expressing GFP-YKT6 were treated with 1 mM LLOMe for 1 h. The lysates were immunoprecipitated with GFP-trap beads and immunoblotted with the indicated antibodies.
(TIF)

**S7 Fig. PACSIN1 is required for clearance of aggresome.** WT and *PACSIN1* KO HeLa cells were treated with 20 μM MG132 for 6 h or 2.5 μg/mL puromycin for 6 h. The lysates were separated to detergent-soluble and detergent-insoluble fraction, and then immunoblotted with the indicated antibodies.
(TIF)

**S1 Movie. PACSIN1 is required for the clearance of α-synuclein in *C. elegans* (Related to Fig 7E).** The locomotion of animals (WT expressing α-synuclein::YFP) at the day 1 stage.
(MP4)

**S2 Movie. PACSIN1 is required for the clearance of α-synuclein in *C. elegans* (Related to Fig 7E).** The locomotion of animals (*sdpn-1* mutants expressing α-synuclein::YFP) at the day 1 stage.
(MP4)

**S3 Movie. PACSIN1 is required for the clearance of α-synuclein in *C. elegans* (Related to Fig 7F).** The locomotion of animals (WT expressing α-synuclein::YFP) at the day 5 stage.
(MP4)

**S4 Movie. PACSIN1 is required for the clearance of α-synuclein in *C. elegans* (Related to Fig 7F).** The locomotion of animals (*sdpn-1* mutants expressing α-synuclein::YFP) at the day 5 stage.
(MP4)

## Acknowledgments

We thank Prof. S. Yamaoka for providing pMRX-IRES-puro and pMRX-IRES-bsr, Prof. BD Grant for providing *sdpn-1*(ok1667) worms, and the *C. elegans* Genetic Center (CGC) at the University of Minnesota for providing the worm strains. We thank Dr. A. Kuma for valuable discussion.

## Author Contributions

**Conceptualization:** Yukako Oe, Tamotsu Yoshimori, Shuhei Nakamura.

**Formal analysis:** Yukako Oe, Keita Kakuda.

**Funding acquisition:** Tamotsu Yoshimori, Shuhei Nakamura.

**Investigation:** Yukako Oe, Keita Kakuda, Shin-ichiro Yoshimura.

**Methodology:** Naohiro Hara, Junya Hasegawa, Seigo Terawaki, Yasuyoshi Kimura, Kensuke Ikenaka, Shiro Suetsugu, Hideki Mochizuki.

**Resources:** Shiro Suetsugu.

**Supervision:** Tamotsu Yoshimori, Shuhei Nakamura.

**Validation:** Yukako Oe, Keita Kakuda.

**Visualization:** Yukako Oe.

**Writing – original draft:** Yukako Oe, Shuhei Nakamura.

**Writing – review & editing:** Yukako Oe, Tamotsu Yoshimori, Shuhei Nakamura.

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
