## [Decision Letter · Decision Letter 0]

19 Jan 2022

Dear Dr Nakamura,

Thank you very much for submitting your Research Article entitled 'PACSIN1 is indispensable for amphisome-lysosome fusion during basal autophagy and lysophagy' to PLOS Genetics.

The manuscript was fully evaluated at the editorial level and by independent peer reviewers. The reviewers appreciated the attention to an important problem, but raised some substantial concerns about the current manuscript. Based on the reviews, we will not be able to accept this version of the manuscript, but we would be willing to review a much-revised version. We cannot, of course, promise publication at that time.

In particular, please focus your changes for an eventual resubmission on improving the clarity of figures and labels, and on addressing the key mechanistic questions and controls raised by reviewers.

If you decide to revise the manuscript for further consideration at PLOS Genetics, please aim to resubmit within the next 60 days, unless it will take extra time to address the concerns of the reviewers, in which case we would appreciate an expected resubmission date by email to plosgenetics@plos.org.

[LINK]

We are sorry that we cannot be more positive about your manuscript at this stage. Please do not hesitate to contact us if you have any concerns or questions.

Yours sincerely,

Javier E. Irazoqui

Associate Editor

PLOS Genetics

Gregory P. Copenhaver

Editor-in-Chief

PLOS Genetics

Reviewer's Responses to Questions

**Comments to the Authors:**

Reviewer #1: Referee #? (Remarks to the Author): PGENETICS-D-21-01574

In this review article, the authors implicate that PACSIN1 plays an important role in the activation of autophagy, particularly in the amphisome-lysosome fusion process. They also claim that PACSIN1 at this time has the task of forming lysophagy by binding to the SNARE complex in autophagy.

Overall, it is thought that they specifically demonstrated the role of PACSN1 specifically for amphisome lysosome fusion based on various experiments. I am satisfied with the amount of trustworthy data presented in this paper. However, some flaws in the paper undermine the study's validity, and these can preclude the publication of this article. But, I can agree to accept this paper if the authors can solve all the problems.

Major concerns:

1. Studies on PACSIN1 have already been conducted in several other papers on the endocytosis recycle together with the endosome and dynamic trafficking of AMPA receptors (AMPARs). In that study, PACSIN1 controls AMPA receptor trafficking in living hippocampal neurons. Since not many studies on PACSIN1 have been published, it is necessary to consider whether there is a correlation between the researcher’s study and the genes already published in the paper mentioned below.

PACSIN1 regulates the dynamics of AMPA receptor trafficking. Widagdo J, Fang H, Jang SE, Anggono V. Sci Rep. 2016 Aug 4;6:31070.

PICK1 interacts with PACSIN to regulate AMPA receptor internalization and cerebellar long-term depression. Anggono V, Koç-Schmitz Y, et al. Proc Natl Acad Sci U S A. 2013 Aug 20;110(34):13976-81.

Native KCC2 interactome reveals PACSIN1 as a critical regulator of synaptic inhibition. Vivek Mahadevan, C Sahara Khademullah, et, al. Elife. 2017 Oct 13;6:e28270.

2. Fig.2. In the PACSIN1 KO data (Fig. 2 AB), it can be seen that the number of cells significantly decreased compared to the Control. Could it be due to other cell death there? apoptosis. necroptosis, pyroptosis, etc... Also, the cathepsin D activity in PACSIN1 KO seems to be all different (Fig. 2B). Some appear to be weaker than control. If it is PACSIN1 KD, it is partially understandable, but if it is PACSIN1 KO, it can be a little problematic. By staining the nucleus with DAPI, it is necessary to show whether the number of cells and the cell status affected by apoptosis (apoptotic body), etc. Be clear about this part.

3. Fig.5. If PACSIN1 binds to both SNARE complexes under lysophagy, why does SNAP29 increase although STX7 binding decreases in Figure 3B? What does this mean? No specific mention is made anywhere.

4. Optional request. Most of the research on the lysophagy environment is common to see in detail by giving a stimulus such as LLOMe (Please refer to the references provided). It needs to be clarified, if possible. (This is the biggest part of the researcher's availability, so if time and circumstances are impossible, you do not have to do it.)

Quantitative proteomics reveals the selectivity of ubiquitin-binding autophagy receptors in the turnover of damaged lysosomes by lysophagy. Vinay V Eapen, Sharan Swarup, Melissa J Hoyer, Joao A Paulo, J Wade Harper. Elife. 2021 Sep 29;10:e72328.

The ubiquitin-conjugating enzyme UBE2QL1 coordinates lysophagy in response to endolysosomal damage. Lisa Koerver

Chrisovalantis Papadopoulos, et al. EMBO Reports (2019)20:e48014

ESCRT-mediated lysosome repair precedes lysophagy and promotes cell survival. Maja Radulovic, Kay O Schink, et al. The EMBO Journal (2018)37:e99753

Minor concerns:

1. No page number. Please insert the page number.

2. Fig. 3A. Statistical data is required if there are several electron microscope data.

3. Fig. 6. For research on Parkin's related mitophagy (autophagy), it is a reasonable experimental method to give a specific stimulus such as CCCP. It is meaningless to see only PACSIN1 KO in the state of giving nothing. Please make this part clear.

12-29-2021

Reviewer #2: In this manuscript, authors found PACSIN1 regulate the fusion between amphisome and lysosome, thus, control basal autophagic degradation and some but not all selective autophagy. The most surprising and important finding of this study is there are two paths for autophagic flow, one for basal autophagy and one for starvation induced autophagy, and these two paths can be distinguished by PACSIN1 dependency. Overall, this is a conceptually novel study backed by solid evidence, I like to suggest a few experiments to further improve this study.

1. Amphiosome can be distinguished from lysosome by lack some of lysosome marker proteins. The accumulation of amphisome in PACSIN1 KO cells need to be better characterized by staining cells with proteins confined to lysosome compartment.

2. Does PACSIN1 also regulate the removal of protein aggregate in mammalian cells?

3. Does F-BAR domain required for amphisome/lysosome fusion? Along this line, does F-BAR domain required for localization of PACSIN1 on autophagosome and lysosome?

4. The figure need to be better labeled.

Reviewer #3: Major comments

It will help understanding if the authors add more details about the quantification of LC3 flux assay and autophagic flux (Fig 1A and B). Given the graphic doesn’t show the increase in LC3-II in PACSIN1 KO.

Minor comments.

-Results section (Loss of PACSIN1 impairs autophagic activity): Please add Supplemental Figure 1 after CRISPR-Cas9 system.

-PACSIN1 KO: Is it a single clone? or the authors isolate multiple independent clones to confirm phenotypes across the clones avoiding an artifact of a single clone. Please clarify this in the new version.

-Fig. 1A: It will help if the authors add the kDa (size) in the western blot for each protein and indicate the LC3-I and II bands.

-Supplemental Fig. 1B: Please correct the size (kDa) of alpha-tubulin to ~50kDa.

-Supplemental Fig. 1C: It will help if the authors add the alpha-tubulin as a loading control.

-Fig. 1B and D: It will help if the authors add the kDa (size) in the western blot for each protein and probe the membrane with alpha tubulin instead show the Ponceaus-S.

-Fig 1C: Did the authors normalize per cell area? If yes, please add the info to the material and methods.

-Fig 1D: it will help if the authors indicate the LC3-I and II bands in the western blot.

-Fig 1E: Why did the authors treat the cell for 8h with BafA1 instead 2h? as they used in Fig1A, B and D. Please clarify this in the new version.

-Fig 2A/B: How did the authors normalize the lysotracker and magic red intensity? Per cell area? It will help if the authors add this information to the manuscript.

-Fig 3A: Please add PACSIN1KO ST into the figure.

-Fig 3B: The authors should keep constant the treatment with BafA1. In Fig 1A, B and D they used 2h of treatment, in Fig. 1E 8h and here in Fig. 3B 6h. It will help if the authors clarify why they used different times of BafA1 treatment.

-Supplemental Fig. 2A: Please add in the legend the cell line used in the experiment.

-Supplemental Fig. 2B. Is it possible to do LAMP1/Cd63 and LC3 staining together? It will improve the conclusion.

-Fig 5A: It will improve the result if the authors add a control such as cells treat with BafA1.

**Have all data underlying the figures and results presented in the manuscript been provided?**

Reviewer #1: None

Reviewer #2: Yes

Reviewer #3: Yes

PLOS authors have the option to publish the peer review history of their article (what does this mean?). If published, this will include your full peer review and any attached files.

Reviewer #1: No

Reviewer #2: No

Reviewer #3: No

---

## [Decision Letter · Decision Letter 1]

19 May 2022

Dear Dr Nakamura,

We are pleased to inform you that your manuscript entitled "PACSIN1 is indispensable for amphisome-lysosome fusion during basal autophagy and subsets of selective autophagy" has been editorially accepted for publication in PLOS Genetics. Congratulations!

Yours sincerely,

Javier E. Irazoqui

Associate Editor

PLOS Genetics

Gregory P. Copenhaver

Editor-in-Chief

PLOS Genetics

Comments from the reviewers (if applicable):

Reviewer's Responses to Questions

**Comments to the Authors:**

Reviewer #1: Thank you for your sincere answer preparation and specific additional experiments, and I look forward to good research in the future.

Reviewer #2: Authors had addressed my queries satisfactorily.

Reviewer #3: In this new version, the authors addressed all the reviewers' comments improving the paper. I am satisfied with this new version and I agree to accept it for publication.

**Have all data underlying the figures and results presented in the manuscript been provided?**

Reviewer #1: Yes

Reviewer #2: Yes

Reviewer #3: Yes

PLOS authors have the option to publish the peer review history of their article (what does this mean?). If published, this will include your full peer review and any attached files.

Reviewer #1: No

Reviewer #2: No

Reviewer #3: No

**Data Deposition**

http://datadryad.org/submit?journalID=pgenetics&manu=PGENETICS-D-21-01574R1

**Press Queries**

---

## [Editor Report · Acceptance letter]

10 Jun 2022

PGENETICS-D-21-01574R1 

PACSIN1 is indispensable for amphisome-lysosome fusion during basal autophagy and subsets of selective autophagy 

Dear Dr Nakamura, 

We are pleased to inform you that your manuscript entitled "PACSIN1 is indispensable for amphisome-lysosome fusion during basal autophagy and subsets of selective autophagy" has been formally accepted for publication in PLOS Genetics! Your manuscript is now with our production department and you will be notified of the publication date in due course.

With kind regards,

Zita Barta

PLOS Genetics

On behalf of:
